# $NO_y$ production, ozone loss and changes in net radiative heating due to energetic particle precipitation in 2002-2010

Miriam Sinnhuber[1], Uwe Berger[2], Bernd Funke[3], Holger Nieder[1], Thomas Reddmann[1], Gabriele Stiller[1], Stefan Versick[1], Thomas von Clarmann[1], and Jan Maik Wissing[4]

[1]Karlsruhe Institute of Technology
[2]Leibniz-Institut für Atmosphärenforschung
[3]Istituto de Astrofisica de Andalucia
[4]University of Osnabrück

*Correspondence to:* Miriam.Sinnhuber@kit.edu

**Abstract.**

We analyze the impact of energetic particle precipitation on the stratospheric nitrogen budget, ozone abundances and net radiative heating using results from three global chemistry-climate models considering solar protons and geomagnetic forcing due to auroral or radiation belt electrons. Two of the models cover the atmosphere up to the lower thermosphere, the source

region of auroral NO production. Geomagnetic forcing in these models is included by prescribed ionization rates. One model reaches up to about 80 km, and geomagnetic forcing is included by applying an upper boundary condition of auroral NO mixing ratios parameterized as a function of geomagnetic activity. Despite the differences in the implementation of the particle effect, the resulting modeled $NO_y$ in the upper mesosphere agrees well between all three models, demonstrating that geomagnetic forcing is represented in a consistent way either by prescribing ionization rates or by prescribing $NO_y$ at the model top.

Compared with observations of stratospheric and mesospheric $NO_y$ from the MIPAS instrument for the years 2002–2010, the model simulations reproduce the spatial pattern and temporal evolution well. However, after strong sudden stratospheric warmings, particle induced $NO_y$ is underestimated by both high-top models, and after the solar proton event in October 2003, $NO_y$ is overestimated by all three models. Model results indicate that the large solar proton event in October 2003 contributed about 1–2 Gmol ($10^9$ mol) $NO_y$ per hemisphere to the stratospheric $NO_y$ budget, while downwelling of auroral $NO_x$ from the

upper mesosphere and lower thermosphere contributes up to 4 Gmol $NO_y$. Accumulation over time leads to a constant particle-induced background of about 0.5–1 Gmol per hemisphere during solar minimum, and up to 2 Gmol per hemisphere during solar maximum. Related negative anomalies of ozone are predicted by the models nearly in every polar winter, ranging from 10–50% during solar maximum to 2–10% during solar minimum. Ozone loss continues throughout polar summer after strong solar proton events in the Southern hemisphere and after large sudden stratospheric warmings in the Northern hemisphere. During

mid-winter, the ozone loss causes a reduction of the infrared radiative cooling, i.e., a positive change of the net radiative heating (effective warming), in agreement with analyses of geomagnetic forcing in stratospheric temperatures which show a warming in the late winter upper stratosphere. In late winter and spring, the sign of the net radiative heating change turns to negative (effective cooling). This spring-time cooling lasts well into summer and continues until the following autumn after large solar proton events in the Southern hemisphere, after sudden stratospheric warmings in the Northern hemisphere.

# 1 Introduction

Energetic particle precipitation is a potential contributor to the solar influence on the middle atmosphere, and has recently been recommended for the first time as a solar forcing parameter for the upcoming CMIP-6 model studies (Matthes et al., 2017). Energetic particles precipitating into the atmosphere lead to the formation of neutral radicals like, e.g., H, OH, N, and NO by reaction chains involving ionization, excitation, and dissociation of the most abundant species, $N_2$ and $O_2$, and subsequent ion chemistry (Sinnhuber et al., 2012). Both $HO_x$ (H, OH) and $NO_x$ (N, NO, $NO_2$, $NO_3$) contribute to catalytic ozone loss in the middle atmosphere, $HO_x$ mainly in the mesosphere (above $\approx 1$ hPa), $NO_x$ mainly in the stratosphere (below $\approx 1$ hPa) (Lary, 1997). Energetic particles come from different sources, mainly from the sun, but also from outside the solar system (Mironova et al., 2015).

Of particular importance for the middle atmosphere are protons from large eruptions of the solar corona, so-called coronal mass ejections, and electrons from high-speed solar wind streams further accelerated in the terrestrial magnetosphere.

Solar coronal mass ejections are sporadic events related to sunspots and the solar magnetic cycle; however, though the events are rare and mainly restricted to the declining phase of the solar maximum, protons are accelerated to energies of tens to hundreds of MeV, and can penetrate directly into the mesosphere and upper stratosphere, and in events with particularly hard energy spectra even down to the lower stratosphere. $NO_x$ increases of up to two orders of magnitude in the upper stratosphere and mesosphere as well as mesospheric ozone loss of more than 80 % has been observed related to strong so-called solar proton events (Jackman et al., 2001, 2005, 2009; López-Puertas et al., 2005; Rohen et al., 2005; Funke et al., 2011). Model studies of these events generally show a good morphological agreement, indicating that the main processes during the solar proton events are reasonably well understood (Jackman et al., 2001; Rohen et al., 2005; Funke et al., 2011).

Energetic electrons are accelerated towards Earth due to magnetic reconnections in the magnetotail during auroral substorms; these electrons then precipitate into the lowermost thermosphere down to 90 km. In geomagnetic storms, radiation belt electrons can be accelerated to energies high enough to precipitate into the mesosphere as well. Aurorae and geomagnetic storms are much more frequent than solar proton events, and though particles do not precipitate as far down into the middle atmosphere, the amount of $NO_x$ formed due to these events likely is much larger, being the main source of the strong increase in NO in the high-latitude lower thermosphere. Variations in the density of $NO_x$ in the mesosphere and lower thermosphere related to geomagnetic activity as a proxy for auroral electron precipitation are reported based on observations, e.g., by (Kirkwood et al., 2015; Hendrickx et al., 2015; Sinnhuber et al., 2016). Mesospheric ozone loss and an increase in mesospheric OH has been observed both related directly to increases in electron fluxes (Verronen et al., 2011; Andersson et al., 2014a, b) and geomagnetic activity (Fytterer et al., 2015b).

$NO_x$ from the high-latitude upper mesosphere and lower thermosphere can propagate downward during polar winter in the large-scale downwelling motions of the winter middle atmosphere. As the photochemical lifetime of $NO_x$ is in the range of weeks to months during polar winter, NOx from the upper mesosphere and lower thermosphere can reach down far into the stratosphere. Enhanced values of mesospheric and stratospheric $NO_x$ attributed to auroral production or geomagnetic activity have been observed sporadically in polar winters for many decades (Solomon et al., 1982; Siskind et al., 2000; Randall et al.,

2007; Sinnhuber et al., 2011). However, these observations were mostly limited to sunlit areas, and thus did not observe deep into polar night. Observations from the Michelson Interferometer for Passive Atmospheric Sounding (MIPAS, Fischer et al. (2008)), which covered also the polar night region, show that $NO_x$ produced by energetic particle precipitation (called EPP $NO_x$ in the following) reaches down to altitudes below 30 km in both hemispheres, in nearly all winters observed (Funke

et al., 2014a). Particularly high values of EPP $NO_x$ are observed in Northern hemisphere late winters after the strong sudden stratospheric warming events in winters 2003/2004, 2005/2006, and 2008/2009 (Randall et al., 2006, 2009; Funke et al., 2014a; Sinnhuber et al., 2014; Funke et al., 2017). These warmings were followed by long-lasting downwelling in the mesosphere and upper stratosphere enabled by a strong polar vortex re-forming after the event. It was shown both from observations (Siskind et al., 2007; Orsolini et al., 2010) and model results (Limpasuvan et al., 2016) that this period of enhanced downwelling was

characterized by the formation of an elevated stratopause in the upper mesosphere.

The so-called EPP indirect effect (Randall et al., 2007) due to downwelling of auroral $NO_x$ into the stratosphere leads to an increase in the catalytic ozone loss in the upper and middle stratosphere progressing from the upper to the lower stratosphere during polar winter. Lower values of ozone in the upper stratosphere are observed during winters characterized by large particle forcing or enhanced values of $NO_y$ (Natarajan et al., 2004; Randall et al., 2005), and downwelling negative ozone anomalies

have been observed for the first time by (Fytterer et al., 2015a; Damiani et al., 2016). However, quantification of the particle induced ozone loss by observations is difficult, because a) MIPAS observations of EPP NOx show that EPP-induced ozone loss must occur in every year, so only relative differences can be obtained from observations; b) stratospheric ozone is quite variable anyway, and c) a much longer timeseries than used by Fytterer et al. (2015a) would be necessary to attribute the observed anomalies clearly to the particle precipitation. Model studies are more suited to study the ozone loss, as models can

do on/off experiments with and without particle precipitation in a clearly defined way.

Such model studies were carried out in the past (Rozanov et al., 2005; Langematz et al., 2005; Marsh et al., 2007; Baum-gaertner et al., 2009; Reddmann et al., 2010; Baumgaertner et al., 2011; Semeniuk et al., 2011; Rozanov et al., 2012), however, in most cases, either only one particular winter or situation was investigated, the EPP NOx input was not well constrained, or a model experiment with freely adaptable dynamics was carried out, making comparison to observations more difficult.

As ozone is one of the key species in the radiative heating of the middle atmosphere, changes in ozone even above the main ozone layer will have an impact on temperatures and dynamics of the middle atmosphere. Analyses of observations using either geomagnetic activity or the hemispheric power index as proxies for particle precipitation suggest that such a coupling between EPP and atmospheric dynamics indeed exists during polar winter, characterized by a warming of the mid-to late winter upper stratosphere at high latitudes (Lu et al., 2008; Seppälä et al., 2013). Analyses of several decades of surface air temperatures

suggest that geomagnetic activity even affects tropospheric weather systems down to the surface in mid-to late winter (Seppälä et al., 2009; Maliniemi et al., 2014).

However, the supposed changes in stratospheric net radiative heating related to EPP $NO_x$ have, to our knowledge, not been analysed in detail, though the general consensus so far is that a net cooling is expected during late winter and spring due to the reduction in upper stratospheric ozone, in contrast, and possibly contradiction, to the observed upper stratospheric warming;

this contradiction generally is explained by a coupling with wave breaking and reflection (Lu et al., 2008; Seppälä et al., 2009).

In this paper we analyze results from three chemistry-climate models considering proton and electron forcing over the period from mid-2002 to mid-2010, e.g, covering 11 polar winters in both hemispheres. $NO_y$[1] in the middle atmosphere from all three models is compared to MIPAS observations to evaluate the model results. The ozone loss at high latitudes in the middle atmosphere is quantified from the difference of model runs with and without particle impact, and changes in the net radiative heating are estimated from these results.

The models used are described in Sec. 2.1, MIPAS data are described in Sec. 2.3. $NO_y$ intercomparison with MIPAS data and the impact of energetic particle precipitation on middle atmosphere $NO_y$ are shown in Sec. 3, ozone intercomparisons with MIPAS data and the quantification of the EPP impact on ozone and stratospheric heating rates based on model results are discussed in Sec. 4.

## 2   Description of models and observations

### 2.1   Models

We use results from three different models in this study which have been used to determine the impact of energetic particle precipitation in the past (e.g., Funke et al. (2011, 2017)) to analyze differences in the model results due to the implementation of the particle impact and the model transport schemes, and to derive a range of possible model results. The models used are 3dCTM (Sinnhuber et al., 2012), KASIMA (Reddmann et al., 2010), and EMAC (Joeckel et al., 2010). The models cover different vertical regimes: 3dCTM and KASIMA cover the altitude region from roughly the tropopause up to the lower thermosphere (called high-top models in the following), the EMAC model covers altitudes from the surface up to the mesopause (called medium-top models in the following). All three models have a detailed description of middle atmosphere chemistry, and use temperatures and wind fields which are relaxed to meteorological analysis data provided by ECMWF in the stratosphere and below. However, the models differ in the treatment of energetic particles on the atmospheric composition, and in the treatment of the impact of non-resolved gravity waves on atmospheric transport. 3dCTM and KASIMA cover the source region of particle-induced $NO_x$ production, and $NO_x$ production is driven by prescribed ionization rates; 3dCTM additionally also considers photoionization. EMAC does not cover the source region of $NO_x$ production, and the indirect effect is considered by prescribing $NO_y$ at the model upper boundary.

KASIMA and EMAC are chemistry-climate models internally calculating temperatures and wind fields, which however are nudged to meteorological analysis data below the stratopause; 3dCTM is a chemistry-transport model driven by prescribed temperatures and windfields. KASIMA and EMAC use standard parameterizations of the gravity wave drag, while in 3dCTM, only resolved gravity waves are considered, restricting the spectrum to wavelengths $\geq$500 km.

The models are described in more detail in the following subsections. In Sec. 2.2, the different model scenarios used are described.

---

[1]$NO_y$: total inorganic nitrogen: $NO_x$ plus $HNO_3$, 2 $N_2O_5$, $ClNO_3$, and $HNO_4$, sometimes also $BrNO_3$, though the contribution is rather minor.

### 2.1.1 3dCTM

The three-dimensional chemistry and transport model (3dCTM) is an advanced version of the 3dCTM described in Sinnhuber et al. (2012). The model is based on a combination of the stratospheric transport model as described in Sinnhuber et al. (2003) and a chemistry and photolysis code adapted from the SLIMCAT model (Chipperfield, 1999). The model operates on isobaric surfaces and reaches from the tropopause to the lower thermosphere (317.00 hPa - $5 \cdot 10^{-6}$ hPa, approximately 10 - 140 km) with a vertical resolution of 1–3 km. The horizontal resolution is $2.5° \times 3.75°$.

Temperatures, densities and wind fields are prescribed using output data from the Leibniz Institute Middle Atmosphere Model LIMA (Berger, 2008). LIMA is nudged to tropospheric and stratospheric data from ECMWF-ERA40 below 45 km which introduce realistic short-term and year-to-year variability. LIMA applies a triangular horizontal grid structure with 41804 grid points in every horizontal layer ($\Delta x \approx \Delta y \approx 110$ km). This allows to resolve the fraction of the large-scale internal gravity waves with horizontal wavelengths of $\geq 500$ km. Temporal LIMA data are made available to 3dCTM every six hours. 3dCTM uses a family approach for neutral gas-phase constituents in the stratosphere at altitudes below the 0.33 hPa level, but not in the mesosphere and lower thermosphere. The impact of energetic particles is considered by prescribed ionization rates for precipitation of electrons, proton and alphas from the AIMOS model (Wissing and Kallenrode, 2009), version v1.6. Photoionization of $N_2$, $O_2$, N and O in the EUV and NO photoionization in the Ly-$\alpha$ band has been included as well. EUV photoionization rates are calculated based on the parameterization of Solomon and Qian (2005). Ionic reactions are not included in the chemistry scheme, but the production of odd nitrogen species as a function of ionization rates and atmospheric state is calculated using the parameterization of Nieder et al. (2014), adapted for photoionization by implementing a dependency on the primary ion composition. The production of $HO_x$ is considered using the parameterization of Solomon et al. (1981), an approach which is widely used and has been validated both in comparison to observations of ozone loss, and in comparison to ion chemistry model results, see, e.g., (Funke et al., 2011; Sinnhuber et al., 2012).

### 2.1.2 KASIMA

The Karlsruhe Simulation of the Middle Atmosphere (KASIMA) is a 3-dimensional mechanistic model of the middle atmosphere solving the primitive equations including middle atmosphere chemistry (Kouker et al., 1999). For the simulations presented here, the model was run on isobaric surfaces from 7 to 120 km with a vertical resolution of 750 m in the stratosphere, gradually increasing to 3.8 km at the upper boundary. The horizontal resolution in the simulation is $\approx 5.4° \times 5.4°$ (T21). The model is coupled to the specific meteorological situations by using the analysed geopotential field at the lower boundary (7 km) and applying analysed vorticity, divergence and temperature fields from ECMWF ERA-interim (Dee et al., 2011) below 1 hPa.

The parameterization of the gravity-wave drag is based on the formulation of Holton (1982). The parameterization has been modified compared to the version described in Kouker et al. (1999) in order to better describe the cross mesopause transport often observed after sudden stratospheric warmings. The spectral distribution of the vertical momentum flux is now described with a Gaussian function of centroid of 7 m/s and a standard deviation of 50 m/s with phase speeds of 0, 20, 40, 60 an 80 m/s.

The filter condition for critical phase speeds has been extended to be applied when the absolute difference between the speeds is less than 10 m/s. The latter condition effectively prevents gravity waves of low phase speed from propagating and breaking in the lower mesosphere. Only gravity waves of higher phase speed then break at higher altitudes, causing an elevated stratopause to build. In addition, the numerical implementation of the vertical diffusion has been re-formulated for better mass conservation according to Schlutow, Becker and Körnich (2014). The model includes a full middle atmosphere chemistry scheme based on a family concept. In the mesosphere, the family members are transported separately. To consider the impact of energetic particle precipitation, proton and electron ionization rates from the AIMOS model (Wissing and Kallenrode, 2009) version v1.6 are prescribed. 1.25 $NO_x$ per ion pair are produced with a partitioning between N and NO of 45% and 55% as described in (Porter et al., 1976; Jackman et al., 2005). 0–2 $HO_x$ per ion pair are formed following (Solomon et al., 1981).

### 2.1.3 EMAC

The ECHAM/MESSy Atmospheric Chemistry (EMAC) model is a numerical chemistry and climate simulation system that includes sub-models describing tropospheric and middle atmosphere processes and their interaction with oceans, land and human influences (Joeckel et al., 2010). It uses the second version of the Modular Earth Submodel System (MESSy2) to link multi-institutional computer codes. The core atmospheric model is the 5th generation European Centre Hamburg general circulation model (ECHAM5, Roeckner et al., 2006). For the present study we used ECHAM5 version 5.3.02 and MESSy version 2.52. The model covers the vertical range from the surface up to 0.01 hPa on 90 (pseudo-) pressure layers with a vertical resolution of about 1 km. The horizontal resolution is $2.8° \times 2.8°$ (T42L90). The model is nudged to ERA-Interim reanalysis data from the surface to 1 hPa with decreasing nudging strength in the transition region in the six levels above. For gravity waves we use the submodel GWAVE which contains the original Hines non-orographic gravity wave routines (Hines, 1997) from ECHAM5 in a modularised structure. For the parameter rmscon (root-mean-square wind-speed at bottom launch level of 642.90 hPa), which controls the dissipation of gravity waves, we use a value of 0.92 m/s. For gas phase reactions the submodel MECCA is used (Sander et al., 2011), and for photolysis the submodel JVAL (Sander et al., 2014). 218 gas phase reactions and 68 photolysis reactions are included. Most of the reaction constants are taken from Sander et al. (2011b). A family concept for $NO_x$ is applied in the whole model domain. Geomagnetic forcing of $NO_y$ in the mesosphere is considered by applying an upper boundary condition (UBC) of $NO_y$, parameterized by the geomagnetic Ap index (newly develope submodel UBCNOX). This applies an online version of the upper boundary condition for the amount of $NO_x$ described in (Funke et al., 2016) and (Matthes et al., 2017). All three parts of the parameterization (background, energetic particles and elevated stratopause events (ESE)) are calculated directly on the model grid. ESEs are detected online by the criteria suggested in Funke et al. (2016) using a threshold value of 53 K for the temperature gradient between 0°–30° N and 70°–90° N at 1 hPa. For the Ap-index values recommended for CMIP6 (Matthes et al., 2017) are used. NO is prescribed in the four highest levels of the model (pressure at midpoint 0.01-0.09 hPa) instead of $NO_y$, and $NO_2$ is set to 0 in those levels to balance $NO_x$ (see (Matthes et al., 2017)). For solar proton events we use the submodel SPE (Baumgaertner et al., 2010), incorporating daily values of precalculated ionization rates as described in (Jackman and Fleming, 2000) which are available updated to 2015 based on observed proton fluxes at *http://solarisheppa.geomar.de/cmip6*. Full ionization rates are applied where the geomagnetic latitude is greater than

**Table 1.** Properties of the model experiments used in this study.

| Models | Experiments | Ionisation rates | EPP $NO_x$ | Period |
|---|---|---|---|---|
| 3dCTM | Base | No ionization | none | 01/1999 – 05/2010 |
| | v1.6 phioniz | Aimos v1.6 $p^+ + e^-$ + photoionization | N/NO variable (Nieder et al., 2014) | 01/1999 – 05/2010 |
| KASIMA | Base | No ionization | none | 09/2002 – 12/2010 |
| | v1.6 | Aimos v1.6 $p^+ + e^-$ | 1.25 $NO_x$/IPR, N / NO constant | 09/2002 – 12/2010 |
| EMAC | Base | No ionization | none | 01/1999 – 03/2012 |
| | UBC | Jackman $p^+$ | $NO_x$ upper boundary | 01/1999 – 03/2012 |

$60°$. For every ion pair produced, 0.55 N and 0.7 NO are formed as suggested by Jackman et al. (2005). Between zero and 2 OH are formed per ion pair based on Solomon et al. (1981) as described in Baumgaertner et al. (2010). Note that effects from the SPE submodel in NO and $NO_2$ are overwritten by the UBCNOX submodel in the four highest model levels.

## 2.2 Model experiments

The model scenarios used in the following are listed with their main properties in Tab. 1. All models carried out model runs over the period of ENVISAT observations, 3dCTM from January 1999 – May 2010, KASIMA from September 2002 to December 2011, and EMAC from 1999 to March 2012. 3dCTM runs are limited by the availability of LIMA data, which when the model runs were set up, were available only until mid-2010.

All models carried out one reference scenario with no particle impact, called Base in the following. Additionally, all models carried out one scenario including full particle forcing as available to the respective model: 3dCTM including protons and electrons from the AIMOS model v1.6 and photoionization (v1.6 phioniz); KASIMA including protons and electrons from the AIMOS model v1.6 (v1.6); and EMAC including protons from the Jackman et al. (2005) database and $NO_y$ upper boundary conditions as a constraint for the EPP indirect effect (UBC, Funke et al. (2016)).

## 2.3 MIPAS

The Michelson Interferometer for Passive Atmospheric Sounding (MIPAS, Fischer et al. (2008)) is a limb-viewing infrared spectometer on the Envisat research satellite. MIPAS measured atmospheric emission from which vertical profiles of temperature and various trace species are inferred. MIPAS provided global coverage in an altitude range from cloud top altitude to about 68 km in its nominal observation mode. The MIPAS measurement period was 2002 to 2012 with a major data gap due to technical problems in 2004. After the interruption of operation the measurement was changed towards inferior spectral but improved spatial resolution but this technical issue is of minor relevance to this study. Data products from the retrieval processor built and operated by the Institute of Meteorology and Climate Research (IMK) at the Karlsruhe Institute of Technology (KIT) in cooperation with the Instituto de Astrofísica de Andalucía (IAA-CSIC) (von Clarmann et al., 2003, 2009) are used. The

following MIPAS data obtained from the nominal observation mode are used in this study: $O_3$, $HNO_3$, $ClONO_2$, $N_2O_4$, $NO_2$, and NO. The data versions used here are documented in von Clarmann et al. (2013). The retrieval of NO and $NO_2$ is described in Funke et al. (2005, 2014a). Updates in the ozone retrieval scheme since von Clarmann et al. (2013) are documented in Laeng et al. (2014).

## 2.4  Characterization of datasets used

$NO_y$ data from all model experiments as well as from MIPAS data are used. $NO_y$ is provided by the models on a daily basis as the sum of N, NO, $NO_2$, $NO_3$, 2 $N_2O_5$, $HONO_2$, $HNO_4$, and $ClONO_2$. KASIMA additionally includes HONO and $BrNO_3$. MIPAS observes the $NO_y$-species NO, $NO_2$, $HONO_2$, $HNO_4$, $N_2O_5$ and $ClONO_2$, and the total observed $NO_y$ is calculated from these. Model results as well as observations are averaged over high latitudes (70–90°N/S) as area-weighted daily averages. For MIPAS, daily averages are derived from both upleg and downleg observations, e.g., for local times of around 10 pm and 10 am. For KASIMA and 3dCTM, model data are put out once per day for the whole model domain at 12:00 UT, and this snapshot is then averaged over the latitude bands. For EMAC, model data are output 2-3 times per day (every ten hours).

Ozone volume mixing ratios in 70–90°N/S are provided in the same way as the $NO_y$ data. Additionally, daily data of the hemispheric total $NO_y$ content are derived from the model results in Sec. 3.4, and changes in solar heating and cooling rates and the energy absorption are calculated from the modeled ozone fields in Sec. 4.4.

## 2.5  Variability of energetic particle precipitation and the dynamical state of the middle atmosphere, 2002–2010

An overview of the particle and dynamical forcing throughout the time-period investigated here is provided in Fig. 1. Shown are the daily sunspot number as a proxy for solar activity, the daily AE index as a proxy for geomagnetic activity, and temperatures in the mid-stratosphere at high latitudes in both hemispheres as a proxy of the dynamical state of the middle atmosphere. Also marked are days with known solar proton events.

The time-period investigated covers solar activity variations of nearly one full 11-year solar cycle. At the beginning of the time period in 2002, solar activity is shortly after its maximum. It then decreases from maximum to minimum from 2003–2006, and reaches a deep and extended minimum with daily sunspot numbers mostly below 10, often zero, in 2008 and 2009. From late 2009 on, sunspot numbers start to rise again, indicating the start of the next solar cycle.

Geomagnetic activity follows solar activity insofar as activity is high at the beginning and end of the time-series, and lowest in 2008 and 2009. However, short, sporadic events of enhanced geomagnetic activity related to geomagnetic storms and auroral substorms occur even during the deep solar minimum in 2008 and 2009. The frequency of sporadic events seems to be quite high throughout the time-period, though the strength of the events – denoted by the magnitude of the disturbance of the geomagnetic field, i.e., the value of the geomagnetic indices – is lower in the solar minimum period. During solar maximum and the transition to solar minimum (2002–2005), the geomagnetic AE index mostly has values above 100 nT, reaching 400–1200 nT in strong geomagnetic storms. From 2006 on, geomagnetic activity falls below 100 nT in quiet days. From 2006 to 2008 and again in 2010, 400 nT is exceeded during stronger geomagnetic storms; during the deepest minimum in 2009, even stronger storms fall below 400 nT.

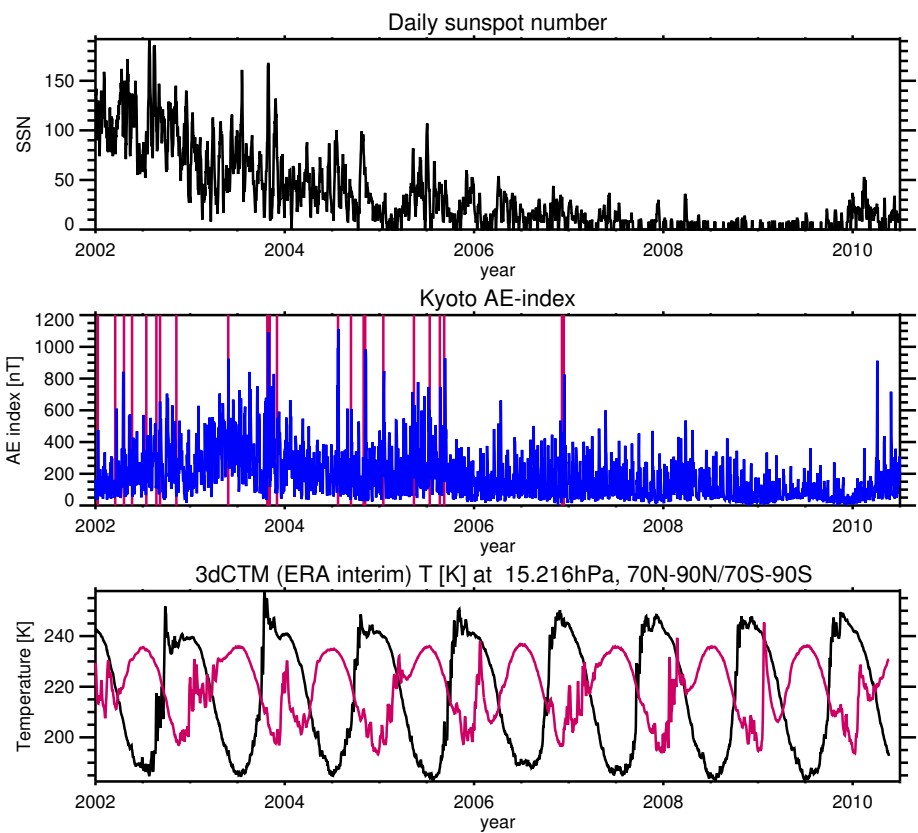

**Figure 1.** Top: Daily sunspot number from *www.ngdc.noaa.gov/stp/spidr.html*, May 2012. Middle: daily mean AE index from the Kyoto database (*http://wdc.kugi.kyoto-u.ac.jp*, February 2016). Red lines denote days of strong solar proton events as given by *http://umbra.nascom.nasa.gov/SEP/*, February 2017 (see text). For events lasting more than one day, only the first day is marked. Bottom: temperature in the mid-stratosphere (15.216 hPa) at high Southern (black, 70–90°S) and high Northern (red, 70–90°N) latitudes. Data are taken from output of the 3dCTM, but originate from ERA-40. The tickmarks here and in all following figures mark the first day of a year.

Days of known solar proton events as provided by *http://umbra.nascom.nasa.gov/SEP/* as mean fluxes of protons with energies larger than 50 MeV are marked as red lines in the middle panel of Fig. 1. Only events where the mean flux is larger than 50 pfu[2] are shown here. Very large events with fluxes of $\geq$ 50 MeV protons larger than 1000 pfu occur on April 21, 2002; October 28–29, 2003; November 02–03, 2003; July 25–26, 2004, January 16–17, 2005; May 14–15, 2005; September 08–11, 2005; and December 06–07, 2006. The strongest event, with mean proton fluxes of 29500 pfu, was the so-called Halloween storms in late October 2003. No events at all are listed in 2007, 2008, and 2009; one smaller event with proton fluxes of 14 pfu occured in August 2010 (not shown here).

Daily mean stratospheric temperatures in high Southern and Northern latitudes are shown in the bottom panel of Fig. 1 for 15.216 hPa (about 25 km). Temperatures are based on ECMWF ERA-40 re-analysis data taken from output of the 3dCTM model (see Sec. 2.1.1). In the Southern hemisphere, stratospheric temperatures show a smooth annual cycle following a more or less sinusoidal behaviour with maxima during polar summer, minima during mid-winter. Small excursions from this behaviour occur in early summer, denoted by short-term increases of stratospheric temperatures of up to 20 K. These are the final warmings denoting the break-down of the meridional circulation in spring. In the Northern hemisphere, a similar annual cycle is observed, though the amplitude is lower – summers are colder, winters warmer. Also, late winter and spring in the Northern hemisphere is dominated by strong excursions from the smooth behaviour of up to 50 K. These are mid-winter sudden stratospheric warmings, strong disturbances of the mean circulation. Sudden stratospheric warmings throughout the time-period observed here are listed, e.g., in (Kuttipurath and Nikulin, 2012; Kishore et al., 2016). Nine warmings are listed from early 2002 to mid-2010, occuring between late December and late February. Sudden stratospheric warmings lasting for more than 10 days occurred in January 2004, January 2006, and January 2009. These warmings were followed by an elevated stratopause and strong and long-lasting mesospheric and upper stratospheric descent (Randall et al., 2006, 2009; Orsolini et al., 2010; Limpasuvan et al., 2016). They will be called *strong sudden stratospheric warmings* in the following. Changes of the temperature structure and dynamics of the middle atmosphere during and after these events have been investigated in a number of studies (e.g., Manney et al. (2005, 2008, 2009a)). In the daily temperatures shown here, these three events are easily distinguished as increases of temperatures by more than 40 K over a few days (more than 50 K in 2009), and a slow recovery to cold winter temperatures in the weeks afterwards. It has been shown that the onset of these events is driven by planetary wave activity (Pancheva et al., 2008, 2009a, b), while downward transport from the thermosphere after the event is driven mainly by non-orographic gravity waves (McLandress et al., 2013).

## 3 Modeled and observed EPP-NO$_y$

In the following, modeled EPP NO$_y$ is investigated in detail. In a first step, model simulations in the upper mesosphere at 0.01 hPa are compared to investigate how the implementation of particle impacts affects the model results (Sec. 3.1). The temporal variation is investigated and compared to MIPAS observations for the whole vertical domain of the MIPAS observations ($\approx$ 10–68 km, e.g., 200 – 0.03 hPa) to evaluate whether the models capture the main features of the EPP impact (see Sec. 3.2).

---

[2]proton flux units: $\frac{protons}{sr\ cm^2\ s}$

Absolute differences between models and observations are discussed to evaluate how well models reproduce the EPP impact quantitatively (Sec. 3.3). In a last step, the total hemispheric amount of EPP $NO_y$ is derived from model results, and compared to previous derivations from observations (Sec. 3.4).

## 3.1 Model-model intercomparison in the upper mesosphere

In Fig. 2, model results are shown at 0.01 hPa ($\approx$ 80 km) respectively, for high latitudes in both hemispheres (first and third panel). 0.01 hPa corresponds to the center of the uppermost layer of the medium-top EMAC model, and also corresponds to an altitude just below the mesopause where thermospheric $NO_y$ enters the middle atmosphere.

At 0.01 hPa, all three models display a consistent behaviour. All show an annual cycle with strong maxima during winter, when $NO_y$ mixing ratios are roughly two orders of magnitude larger than during polar summer. All three models show an additional daily variability of $NO_y$ related to strong, sporadic events in particle fluxes (3dCTM, KASIMA) respectively geomagnetic activity (EMAC). This day-to-day variability is reproduced consistently by all three models during polar winter. During polar summer it is missing in EMAC as the Ap-dependent contribution in the UBC parameterization is much smaller than the background $NO_y$ contribution during summer. Absolute values of $NO_y$ at 0.01 hPa do not show systematic differences between models in the Southern hemisphere, but in the Northern hemisphere, $NO_y$ from EMAC is higher than $NO_y$ from 3dCTM and KASIMA throughout most winters.

The relative difference between $NO_y$ from EMAC and from the other two models is also shown in Fig. 2 (second and fourth panel). The relative differences confirm that on average, values during Southern hemisphere winter are consistent in all three models, while in Northern hemisphere winters, EMAC shows systematically higher values than the other two models. Another systematic feature is observed during polar winter: The relative differences show a kind of U-shape during most winters, with more negative values (EMAC larger than other models) during mid-winter, more positive values (EMAC lower than other models) during early and late winter. 3dCTM and KASIMA on average agree very well with each other. However, in the Southern hemisphere, there are two winters in which $NO_y$ values in 3dCTM and KASIMA are distincly different, winter 2003 (lower values in 3dCTM) and winter 2009 (lower values in KASIMA). In the Northern hemisphere, 3dCTM and KASIMA appear to agree on average in all winters, with the exception of periods probably related to sudden stratospheric warmings (early 2004, early 2006, early 2008, and early 2009); during these periods, 3dCTM underestimates $NO_y$ compared to KASIMA.

To summarize, at the source region of particle forcings in the upper mesosphere (0.01 hPa), all three models show a reasonable, consistent behaviour. We conclude that the upper boundary condition in a medium-top model, or prescribed ionization rates in the mesosphere and lower thermosphere from the AIMOS model (version 1.6) in a high-top model, lead to a generally consistent description of particle-induced $NO_y$ in the upper mesosphere. However, small systematic differences between the model using the upper boundary condition (EMAC) and the models using AIMOS data (3dCTM, KASIMA) indicates possible systematic differences in $NO_y$ related to the use of the upper boundary condition and the AIMOS rates between hemispheres, and in the annual variation. Sporadic differences between 3dCTM and KASIMA results are likely due to dynamical effects, in particular during the strong sudden stratospheric warmings in Northern hemisphere early 2004, early 2006, early 2008, and early 2009 when 3dCTM underestimates $NO_y$ compared to KASIMA.

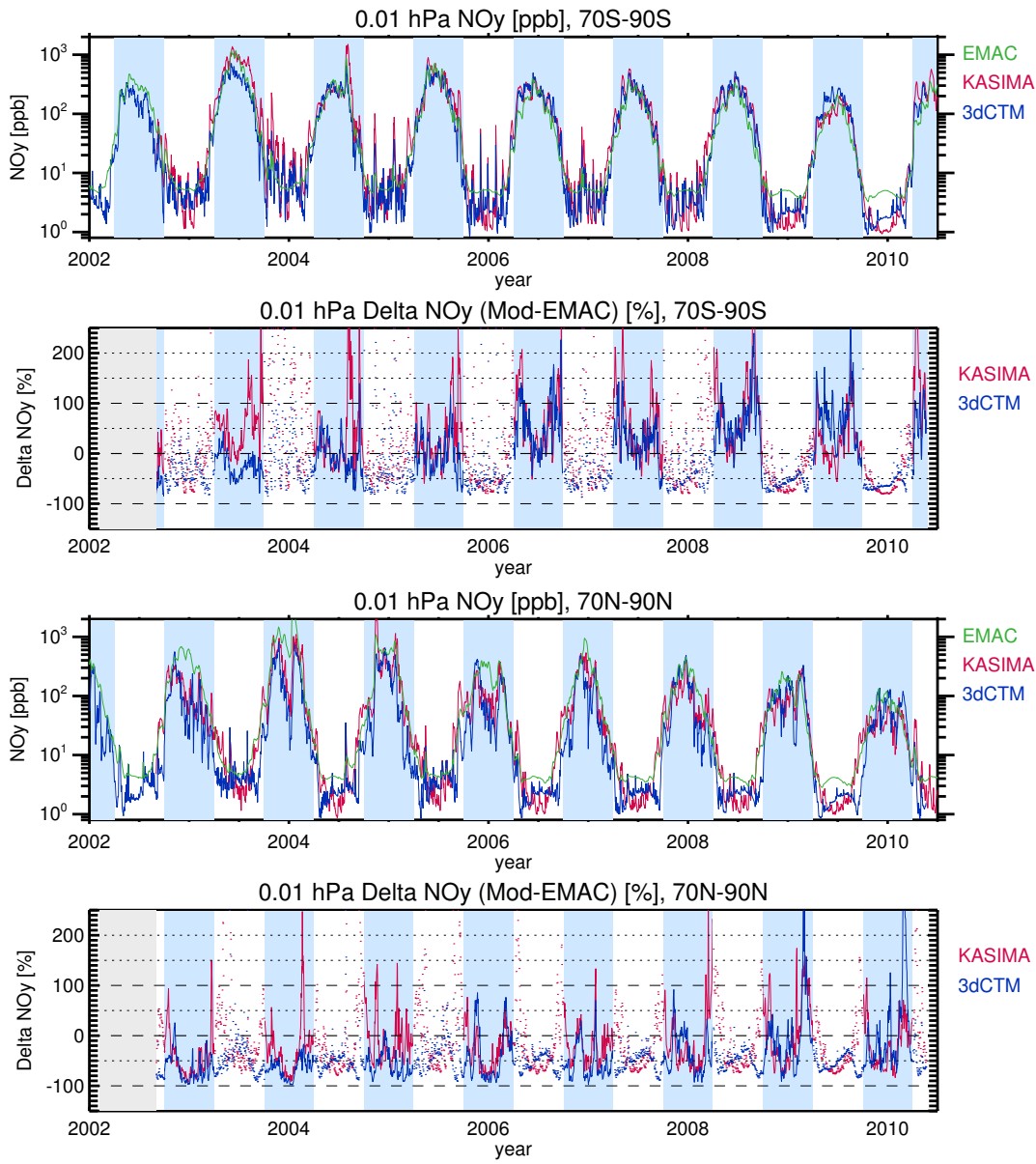

**Figure 2.** Temporal evolution of the volume mixing ratio of $NO_y$ (ppb) from all three models at high Southern and Northern latitudes (70–90°S/N), at 0.01 hPa. Also shown are relative differences from KASIMA and 3dCTM to EMAC results (%). Light blue shading denotes winter in the Southern/Northern hemisphere (second and third quarter of each year / first and forth quarter of each year). The gray shading in the difference plots denote the period at the start where KASIMA results were not yet available. To highlight differences in polar winter, differences in summer are shown as dots, while differences in winter are shown as solid lines. KASIMA: red, 3dCTM: blue, EMAC: green.

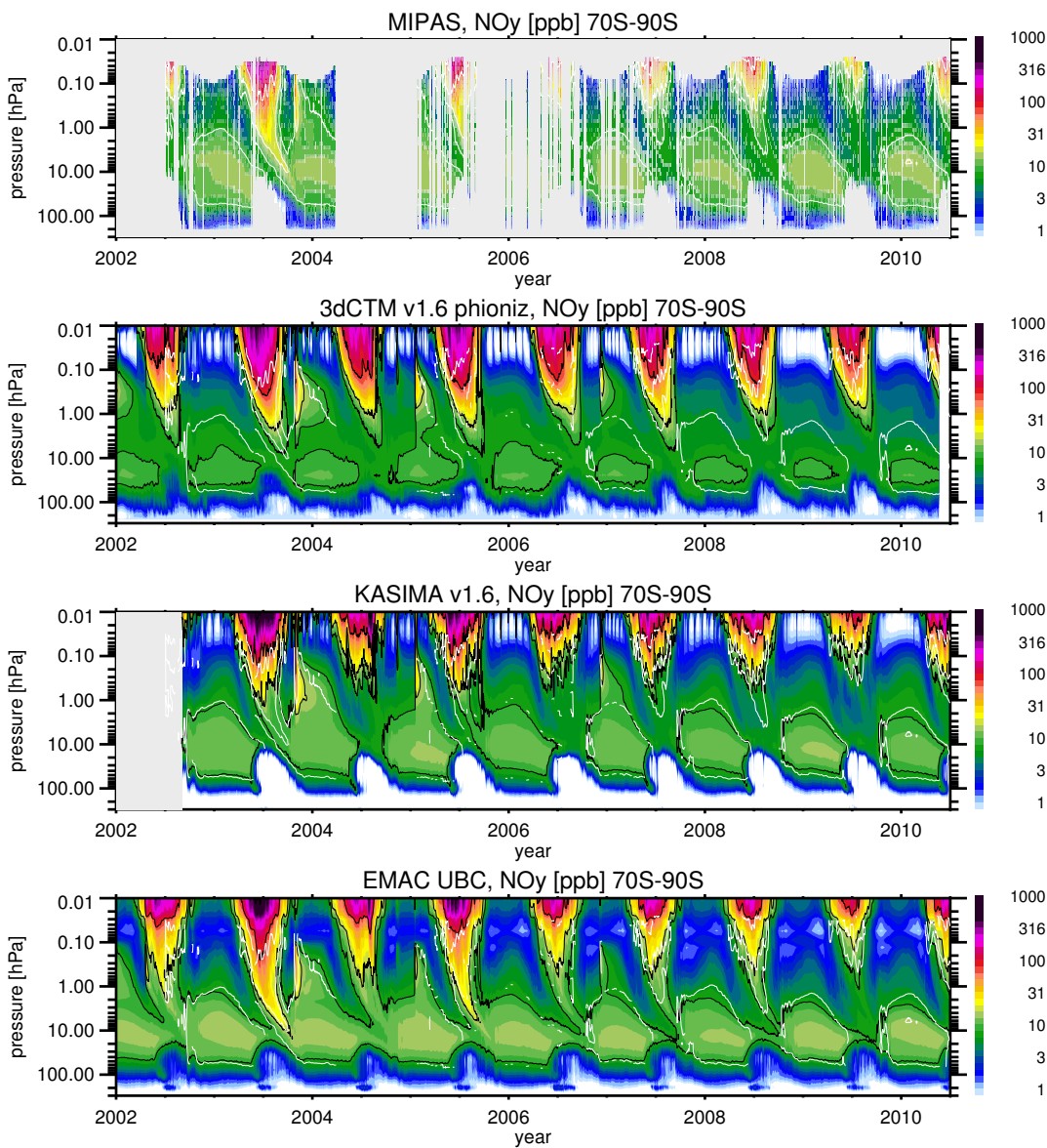

**Figure 3.** Southern hemispheric temporal evolution of the volume mixing ratio of $NO_y$ in all three models and MIPAS measurements at the full common altitude range. Top: MIPAS, second: 3dCTM, third: KASIMA, bottom: EMAC. The white contour lines refer to the respective MIPAS values (at 10, 20, 100, and 1000 ppb) smoothed over 7 days for clarity. Black are the respective model contours (not smoothed). The gray shaded areas in the first panel are periods without MIPAS data, in the third panel, periods before the start of the KASIMA model run.

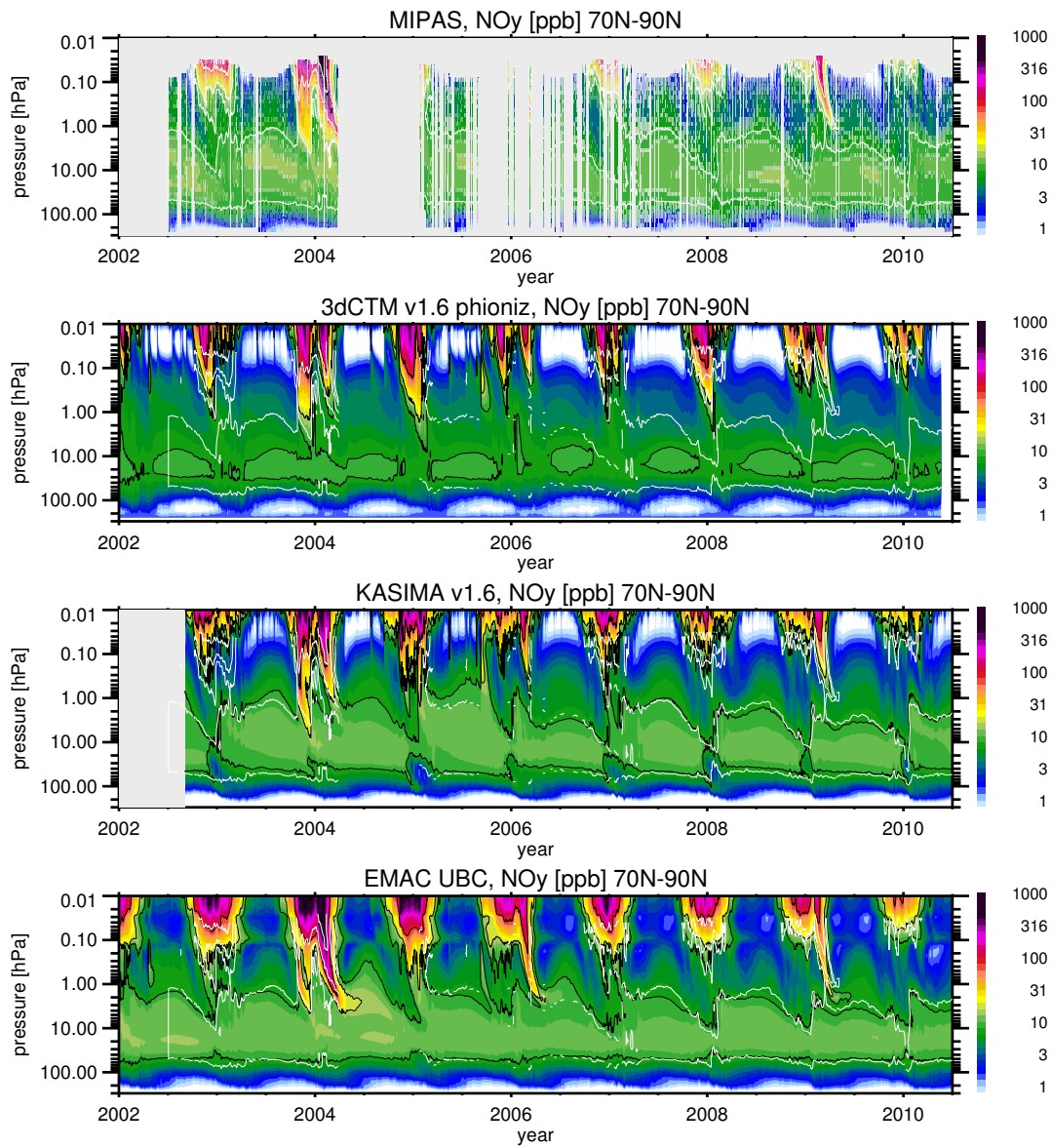

**Figure 4.** Northern hemispheric temporal evolution of the volume mixing ratio of $NO_y$ in all three models and MIPAS measurements at the full common altitude range. Top: MIPAS, second: 3dCTM, third: KASIMA, bottom: EMAC. The white contour lines refer to the respective MIPAS values (at 10, 20, 100, and 1000 ppb) smoothed over 7 days for clarity. Black are the respective model contours (not smoothed). The gray shaded areas in the first panel are periods without MIPAS data, in the third panel, the period before the start of the KASIMA model run.

## 3.2 Variations in the temporal-spatial domain

MIPAS data and model results at high Southern and Northern latitudes are shown in Figs. 3 and 4. White contour lines are MIPAS isolines of 10, 20, 100, and 1000 ppb smoothed over 7 days for clarity, and black contour lines are the same (unsmoothed) isolines for the models. In the MIPAS data, the solar proton event of October/November 2003 (the Halloween storm) is clearly visible as an enhancement in mesospheric ($\leq$1 hPa) $NO_y$ in both hemispheres. Additionally, downwelling of $NO_y$ from above the top altitude covered by MIPAS data (above the mid-mesosphere) is observed in every winter for which MIPAS data are available, also in both hemispheres. The same general structures are simulated by all three models. Additionally, the models also show responses to three more solar proton events (January 2005 and December 2006 in the Southern hemisphere, and September 2005 in the Northern hemisphere) which are not covered by MIPAS observations due to data-gaps. KASIMA and 3dCTM also clearly show a large range of more minor, sporadic events in the upper mesosphere in most polar summers from 2002 – 2008. These occur above the altitude range where MIPAS observations are available during polar summers, and are likely due to mid-energy electron precipitation as provided by the AIMOS model. They are not predicted by EMAC because direct ionization due to electron precipitation is not considered in this model.

While the EPP indirect effect is seen in both observations and model results in all polar winters, some discrepancies are observed in the amount of $NO_y$ transported down into the stratosphere, and in the downward speed and vertical coverage of the EPP signal, both between the different models, and between (all) models, and observations. These differences will be discussed in detail in the following subsection.

## 3.3 Quantification of model-observation differences

For a more quantitative comparison of models and observations, model results from all three models have been interpolated onto the pressure grid of the MIPAS observations for high latitudes (70–90°) in both hemispheres. Only days and altitudes where MIPAS data are available are considered, and the absolute differences in $NO_y$ are shown in Fig. 5 for the Southern, Fig. 6 for the Northern hemisphere. Contours of MIPAS data smoothed over 7 days at 10, 20, and 3000 ppb are overlayed for clarity. These contours are chosen because they envelope the EPP $NO_y$ signal observed by MIPAS in the upper stratosphere and lower mesosphere (10–0.1 hPa) in many winters quite well, in particular in SH winters 2003, 2007, 2008, and 2009, and in NH winters 2002-2003, 2003-2004 and 2008-2009.

In the source region of the particle precipitation, the upper mesosphere above 0.1 hPa, no clear picture emerges. 3dCTM overestimates $NO_y$ in the Southern hemisphere in this region in mid-and late winter, but sometimes underestimates $NO_y$ in early winter. For KASIMA and EMAC, periods of overestimation and underestimation vary throughout most winters. In the Northern hemisphere, 3dCTM overestimates $NO_y$ in the upper mesosphere in early to mid-winter 2007–2008, 2008–2009, and 2009–2010, but underestimates $NO_y$ in most late winters, and throughout winter 2002–2003 and 2003–2004. KASIMA underestimates $NO_y$ compared to MIPAS in the uppermost mesosphere in all Northern hemisphere winters, while in EMAC, again periods of over- and underestimation are observed in all winters.

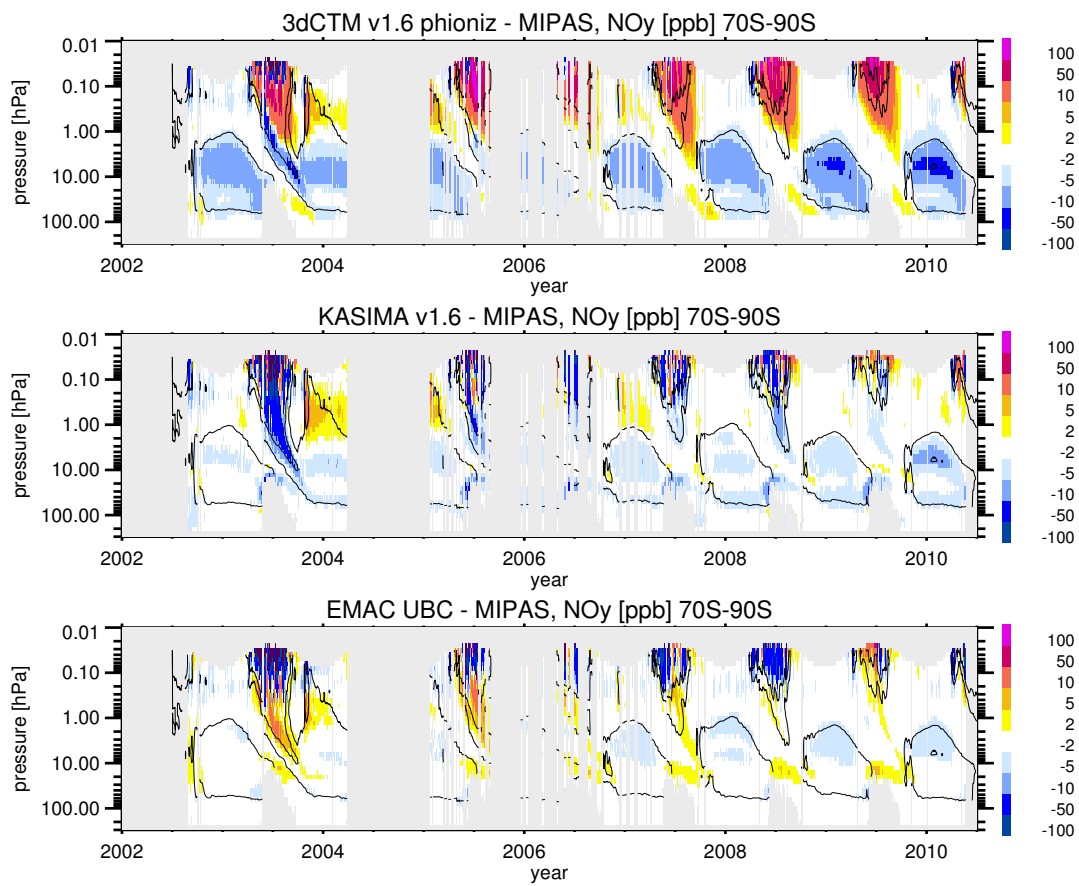

**Figure 5.** Absolute difference in $NO_y$ (ppb) between models and MIPAS observations in high Southern (70–90°S) latitudes. Model data have been interpolated to the vertical grid of the MIPAS data, and only days and altitudes are shown where MIPAS data are available, and fulfill the averaging kernel criterion. From top to bottom: 3dCTM v1.6 phioniz, KASIMA v1.6, and EMAC UBC. The black lines show the 10, 20, and 3000 ppb contours of the MIPAS data, smoothed with a 7-day running mean for clarity. The gray shaded areas are periods without MIPAS data, in the second panel also the period before the start of the KASIMA model run.

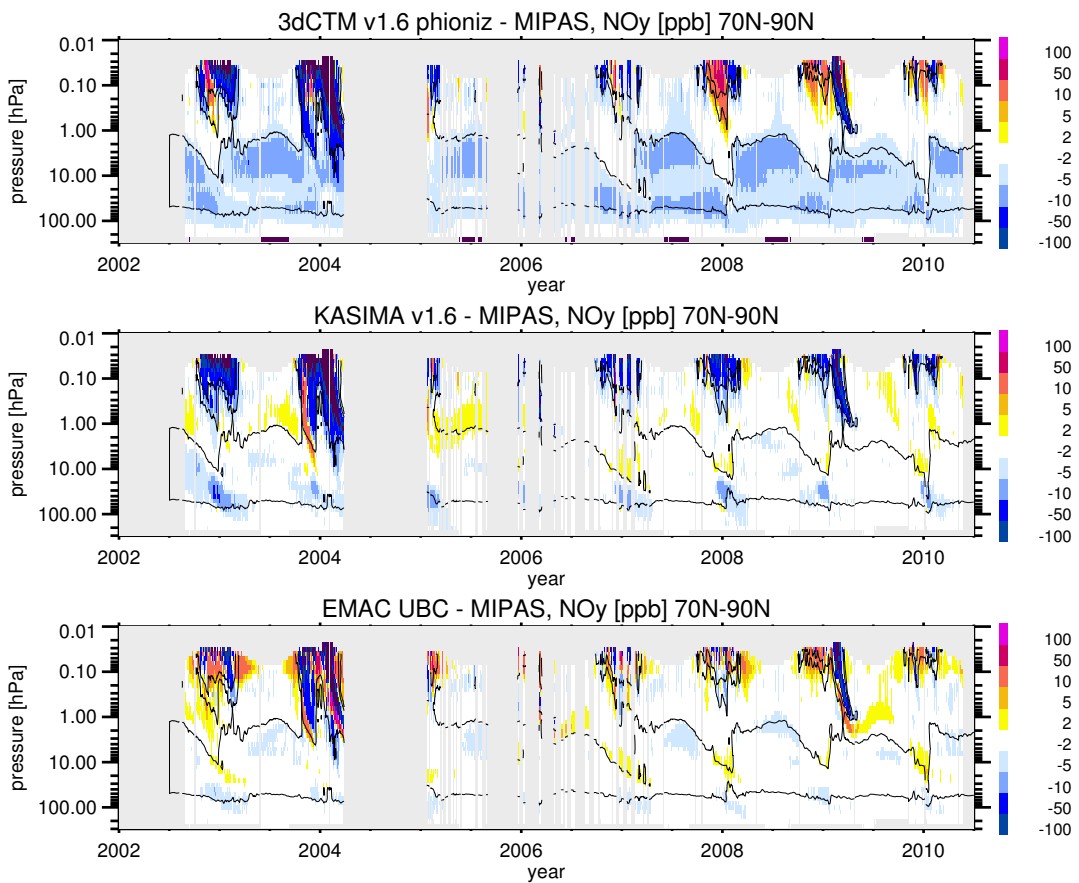

**Figure 6.** Absolute difference in $NO_y$ (ppb) between models and MIPAS observations in high Northern (70–90°N) latitudes. Model data have been interpolated to the vertical grid of the MIPAS data, and only days and altitudes are shown where MIPAS data are available, and fulfill the averaging kernel criterion. From top to bottom: 3dCTM v1.6 phioniz, KASIMA v1.6, and EMAC UBC. The black lines show the 10, 20, and 3000 ppb contours of the MIPAS data, smoothed with a 7-day running mean for clarity. The gray shaded areas are periods without MIPAS data, in the second panel also the period before the start of the KASIMA model run.

Despite the strong similarities of the modeled $NO_y$ in the upper mesosphere during winters discussed in the previous section, the indirect effect in the upper stratosphere to mid-mesosphere (10–0.1 hPa) is captured rather differently by the three models.

- In 3dCTM, the indirect effect is overestimated throughout all polar winters in the Southern hemisphere mesosphere. However, the timing of the downwelling from the mesosphere to the stratosphere is delayed compared to MIPAS observations, which leads to an underestimation of $NO_y$ in the early to mid-winter upper stratosphere in particular in winter 2003. In the Northern hemisphere, the indirect effect is underestimated strongly in winter 2003/2004, but overestimated in winter 2007-2008 and in early winter 2008-2009.

- In KASIMA, the indirect effect in the upper stratosphere to mid-mesosphere is underestimated in all winters and both hemispheres. However, the timing of the downwelling seems to be captured quite well.

- In EMAC, the indirect effect in the upper stratosphere to mid-mesosphere is overestimated in nearly all winters and both hemispheres. The timing of the downwelling $NO_y$ suggests that downwelling is too fast in the upper stratosphere / lower mesosphere.

- 3dCTM and KASIMA strongly underestimate the indirect effect after the sudden stratospheric warmings in early 2004 and early 2009 by more than 100 ppb. The speed of downward transport after the warming is too fast in EMAC: the EPP $NO_y$ signal reaches the stratosphere too quickly, and proceeds to too low altitudes. However, the amount of $NO_y$ transported down after the warming seems to be captured reasonably well. This is consistent with results of the EMAC model for Northern hemisphere winter 2008/2009 as shown in Funke et al. (2017).

These results suggest that in 3dCTM, transport through the winter mesosphere is restricted particularly in the Southern hemisphere, so that $NO_y$ accumulates there throughout the winter. In the Northern hemisphere, a similar overestimation in the upper stratosphere and lower mesosphere is observed in early and mid-winter from 2007–2010, but not in the early years. In contrast, downward transport in EMAC through the lower mesosphere in early winter is too fast, leading to too low values in the mesosphere, too high values in the upper stratosphere.

Only one strong solar proton event is captured by MIPAS observations during this time-period (October 2003). In the Southern hemisphere, $NO_y$ during and after this event is overestimated by the models in the lower mesosphere (1–0.1 hPa), indicating an overestimation of proton ionization rates there. This is shown consistently in all three models. This might indicate either a problem of the proton flux data used, or of the photochemical lifetime of $NO_y$ in the models after the solar proton event. However, it is probably not due to the calculation of the ionization rates, as different models are used by 3dCTM and KASIMA (AIMOS) respectively EMAC (Jackman rates). The impact of the different assumptions in ionization models (including AIMOS and Jackman rates) on $NO_y$ and Ozone is discussed, e.g., in Wissing et al. (2016). This overestimation continues well into the following polar summer. In the Northern hemisphere, modeled $NO_y$ is overestimated below $\approx 0.2$ hPa by KASIMA and EMAC, but underestimated above. The underestimation in the mesosphere seems to be due to an underestimation of the indirect effect, while the overestimation at altitudes below 0.2 hPa has already been discussed in a detailed model-measurement intercomparison of this event (Funke et al., 2011).

In the lower stratosphere below 20 hPa, positive differences occur during mid to late winter occasionally in all models, in particular in the Southern hemisphere, and strongest in 3dCTM. These are likely due to the models' representation of the formation of polar stratospheric clouds in the cold polar vortex, and their subsequent sedimentation out of the stratosphere (denitrification).

3dCTM underestimates $NO_y$ in the polar summer stratosphere by 3–30 ppb. This is observed in both hemispheres, and in all polar summers. KASIMA and EMAC agree much better with observations in the polar summer stratosphere, though a small underestimation of $NO_y$ compared to MIPAS data is observed in these models also during some summers. However, the underestimation of stratospheric summer-time $NO_y$ in 3dCTM is likely connected to background $NO_y$, not to particle precipitation, and will not be discussed further here.

To summarize, the EPP indirect effect in $NO_y$ in the upper stratosphere and lower mesosphere is captured quite differently by all three models depending on the speed of the downward transport in the lower mesosphere and upper stratosphere, which ultimately depends on the different treatment of gravity wave drag in the models. 3dCTM and KASIMA strongly underestimate the indirect effect in $NO_y$ after the strong sudden stratospheric warmings in Northern hemisphere winters 2003–2004 and 2008–2009. This is consistent with results of a dedicated model-measurement intercomparison involving three high-top and five

medium-top models investigating the January 2009 SSW (Funke et al., 2017). The impact of the warming is better represented in EMAC, which however transports $NO_y$ down too fast after the warming. It should be noted that the downward transport is even faster than in the EMAC version shown in (Funke et al., 2017) due to a different setting of the gravity wave drag scheme. Only one strong solar proton event was observed by MIPAS during this time-period; the impact of this event was overestimated by all three models in the Southern hemisphere.

Thus, when the impact of particle precipitation on total $NO_y$, stratospheric ozone loss, and net radiative heating is determined from these models, it should be kept in mind that the indirect effect during Southern hemispheric winters and Northern hemispheric dynamically quiet winters (i.e., winters without strong sudden stratospheric warmings) will likely be underestimated by KASIMA, but will likely be overestimated by 3dCTM and EMAC. However, after sudden stratospheric warmings the impact of the indirect effect will likely be underestimated by 3dCTM and KASIMA, but represented reasonably well by

EMAC. The impact of large solar proton events might be overestimated, though it should be pointed out that this assumption is based on observations of one strong solar proton event only.

### 3.4   Total $NO_y$

The daily total amount of $NO_y$ in each hemisphere is calculated from the model results as follows: In a first step, the total column amount is calculated for daily zonal averages for each model run on the native latitude grid of the respective model.

This is calculated for each model run over the vertical range from 200 hPa (roughly the lower boundary of 3dCTM) to 0.01 hPa (the upper boundary of EMAC). An additional calculation is carried out for the vertical range from 9–0.01 hPa covering a vertical range not affected by denitrification in the Antarctic ozone hole in any winter (compare to Figure 3). The area of each latitude bin is then calculated for each model grid, and the daily hemispheric amount is derived by adding up total $NO_y$ amounts in each latitude bin separately for the Northern and Southern hemisphere, and for each model scenario. The EPP-$NO_y$ amount

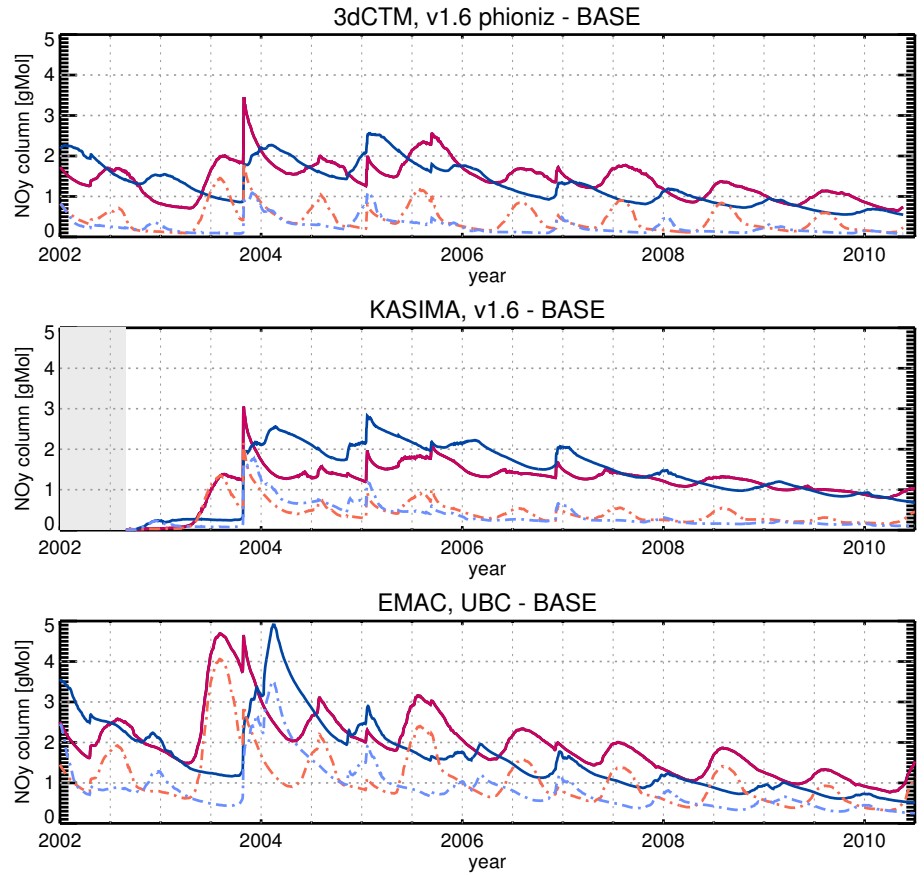

**Figure 7.** Total EPP-NO$_y$ (Gmol) in the middle atmosphere as the difference between model runs with and without particle forcings. From top to bottom: 3dCTM, KASIMA, and EMAC. Red: hemispheric amount in the Southern hemisphere, blue: hemispheric amount in the Northern hemisphere; solid lines, dark colors: 200–0.01 hPa (lower stratosphere to mesopause); dashed lines, light colors: 9–0.01 hPa (mid-stratosphere to mesopause). The gray shading in the second panel denotes the period at the start where KASIMA results were not yet available.

is derived as the difference between a model run with full particle forcing (3dCTM v1.6 phioniz, KASIMA v1.6, EMAC UBC) to the respective Base model run without particle forcing.

Results of the total EPP-$NO_y$ amount are shown for all three models in Figure 7. All three models show a similar behaviour. The main features are:

- In both hemispheres, EPP-$NO_y$ shows values between 0.5 and 5 Gmol, with a distinct annual variability favouring the winter periods, and sporadic, short-lived increases of up to 2 Gmol per hemisphere related to strong solar proton events.

- The total amount of EPP-$NO_y$ is strongest in the winters of the transition from solar maximum to minimum: 2003 to 2005 in the Southern hemisphere, 2003-2004 and 2004-2005 in the Northern hemisphere. The very high values in Southern hemisphere winter 2003 and Northern hemisphere winter 2004-2005 seem to be due to a combination of the indirect effect transporting $NO_y$ down into the stratosphere, and a strong solar proton event in mid-to late winter (October 2003 and January 2005). The impact of the October 2003 SPE is more pronounced in 3dCTM and KASIMA than in EMAC, where the maximum $NO_y$ values are reached already before the SPE. In Northern hemisphere winter 2003-2004, the October 2003 solar proton event occurred in early winter. The very high values in Northern hemisphere winter 2003-2004 seem to be due to a combination of the large solar proton event in early winter, and the sudden stratospheric warming in late winter. However, very different values are predicted by the models for the impact of the sudden stratospheric warming: an increase of about 2 Gmol in EMAC, half a Gmol in KASIMA, no distinctive increase in 3dCTM.

- EPP-$NO_y$ is enhanced over the whole model period by more than 0.5 Gmol in both emispheres. This indicates that once $NO_y$ has reached the mid-stratosphere, its effective atmospheric lifetime is rather longer than one year. EPP-$NO_y$ accumulates over the solar maximum years, and does not drop to zero before the next maximum starts. This accumulation effect is emphasized by the very low values displayed by KASIMA at the beginning of the KASIMA model period in mid-2002.

- Highest values of EPP-$NO_y$ are predicted by EMAC, lowest by KASIMA, in agreement with the results of the comparison to stratospheric $NO_y$ from MIPAS shown in the previous section.

- The annual variation is less pronounced in the Southern hemisphere $NO_y$ from KASIMA. This might be due to the stronger (and apparently, more realistic) denitrification in KASIMA compared to the other two models: Denitrification redistributes $NO_y$ in the lowermost stratosphere very efficiently, taking $NO_y$ out of the gas-phase and sedimenting it out of the middle atmosphere completely. This becomes evident by comparing to the EPP-$NO_y$ from altitudes above the vertical range where denitrification occurs (9 hPa): this shows a comparable annual variation in all three models.

To summarize, energetic particle precipitation provides a nearly constant background of EPP-$NO_y$ in both hemispheres, from a few tenth of Gmol during solar minimum, to 1–2 Gmol during solar maximum. Superposed on this background is a distinct annual cycle with higher values during polar winter due to the EPP indirect effect, that is, downwelling of particle induced $NO_y$ probably originating in the aurora during polar winter at high latitudes. Additionally, there are sporadic increases

**Table 2.** Comparison of modeled EPP $NO_y$ (Gmol) with data derived from MIPAS Funke et al. (2014b). For the models, the difference between the highest value of this winter minus the lowest value of the preceding summer is given; in brackets, the maximal value of the winter is given.

| Winter | Mipas | 3dCTM | KASIMA | EMAC |
|---|---|---|---|---|
| NH 2002/2003 | 0.51 | 0.29 (1.55) | 0.28 (0.28) | 0.62 (2.24) |
| SH 2003 | 2.5 | 2.74 (3.45) | 3.04 (3.05) | 3.20 (4.69) |
| NH 2003/2004 | 3.17 | 1.41 (2.27) | 2.31 (2.56) | 3.74 (4.91) |
| SH 2004 | - | 0.45 (2.00) | 0.33 (1.56) | 1.06 (3.10) |
| NH 2004/2005 | 1.19 | 1.13 (2.56) | 1.10 (2.82) | 0.95 (2.90) |
| SH 2005 | 1.5 | 1.29 (2.55) | 0.90 (2.09) | 1.36 (3.16) |
| NH 2005/2006 | 0.45 | 0.17 (1.77) | 0.26 (2.23) | 0.40 (1.96) |
| SH 2006 | 0.69 | 0.35 (1.69) | 0.12 (1.43) | 0.76 (2.33) |
| NH 2006/2007 | 0.5 | 0.44 (1.34) | 0.57 (2.06) | 0.65 (1.77) |
| SH 2007 | 0.7 | 0.56 (1.77) | 0.17 (1.43) | 0.57 (2.01) |
| NH 2007/2008 | 0.39 | 0.38 (1.19) | 0.18 (1.48) | 0.37 (1.24) |
| SH 2008 | 0.9 | 0.51 (1.37) | 0.24 (1.32) | 0.81 (1.86) |
| NH 2008/2009 | 0.18 | 0.23 (0.95) | 0.25 (1.20) | 0.28 (1.01) |
| SH 2009 | 0.51 | 0.36 (1.14) | 0.02 (1.007) | 0.36 (1.33) |
| NH 2009/2010 | 0.11 | – | | |

of more than 1 Gmol per event due to strong solar proton events, and in the Northern hemisphere, also due to strong sudden stratospheric warmings.

Strongest $NO_y$ increases are simulated by the models for the October 2003 solar proton event: between 0.9 Gmol per hemisphere (3dCTM in the Northern hemisphere, EMAC in the Southern hemisphere) and 1.8 Gmol per hemisphere (KASIMA in the Southern hemisphere). Increases of several tenths of Gmol are predicted for the solar proton events in January 2005, July 2004, and December 2006. This is in the same order of magnitude as previous estimates of hemispheric $NO_y$ increases for solar proton events in November 1960 (1.1 Gmol, Crutzen (1975)), September 1966 (0.34 Gmol, Crutzen (1975)) and the October 2003 event (0.75–2.82 Gmol, Jackman et al. (2005, 2009); Reddmann et al. (2010)), but lower than estimates for the August 1972 event (2.98–3.40 Gmol, Crutzen (1975); Jackman et al. (2005)) and the October 1989 event (5.56–6.97 gMol per hemisphere, Vitt and Jackman (1996); Jackman et al. (2005)).

The winter-time increase due to the indirect effect is in the range of a few tenth of Gmol in solar minimum, to up to 1.5–3.7 Gmol in the transition from declining phase of the solar maximum. These values are higher than in previous studies summarized in Sinnhuber et al. (2012), which provide a range of zero to 1.5 Gmol per winter based on HALOE (Siskind et al., 2000; Randall et al., 2007) observations, but are in good agreement with studies based on MIPAS data (Funke et al., 2005; Reddmann et al., 2010; Funke et al., 2014b). In Funke et al. (2014b), the wintertime increase in EPP $NO_y$ due to the

indirect effect is derived from MIPAS observations for every Northern hemisphere and Southern hemisphere winter covered by MIPAS observations (mid-2002–early 2012). To make these observations directly comparable to our model results, the increase in every Northern and Southern hemisphere winter is derived from the model results by subtracting the lowest value in the preceding summer. Results for all winters and all model runs are summarized together with the MIPAS observations (Funke et al. (2014b), their Table 1) in Tab. 2. Observations and model results are generally in good agreement, with values of more than 1 Gmol from Southern hemisphere winter 2003 to Southern hemisphere winter 2005 (note that there are no MIPAS observations during Southern hemisphere winter 2004), and values lower than 1 Gmol at the beginning and end of the time-period, in Northern hemisphere winter 2002–2003 and after Southern hemisphere winter 2005. In years with low EPP $NO_y$, 3dCTM and KASIMA are more likely to underestimate EPP $NO_y$, while in EMAC, EPP $NO_y$ is more likely to be in agreement to, or higher than, observed EPP $NO_y$. Highest values of EPP $NO_y$ of more than 3.17 Gmol are observed by MIPAS in Northern hemisphere winter 2003–2004. EPP $NO_y$ in this winter is underestimated by 3dCTM (1.41 Gmol) and KASIMA (2.31 Gmol), but overestimated by EMAC (3.74 Gmol).

## 4    Ozone intercomparison, quantification of ozone loss and net radiative heating change

In the following, modeled ozone is compared to MIPAS observations, and the particle impact on stratospheric ozone and net radiative heating will be quantified from model results by comparing the model runs with full particle forcing (3dCTM v1.6 phioniz, KASIMA v1.6, and EMAC UBC) to the respective Base model scenarios. Changes in ozone are discussed in Sec. 4.3, changes in radiative heating and cooling rates are derived and discussed in Sec. 4.4.

### 4.1    Comparison of modeled and observed ozone fields

In Figs 8 and 9, a comparison of the model results including full particle forcing (3dCTM v1.6 phioniz, KASIMA v1.6 and EMAC UBC) to MIPAS observations is shown at high latitudes (70–90°) in both hemispheres. Mixing ratios of the MIPAS ozone fields are shown as well as the difference between the model and the observations where MIPAS data are available. MIPAS data have been restricted to the vertical range of the limb scans, i.e., below 68 km, though data are retrieved above. The 0.5 ppm and 1 ppm contour lines of the model fields are shown as a reference to how well the main features of the temporal and vertical variation of ozone are captured. Ozone is characterized by a winter-time maximum in the upper mesosphere (above ≈0.4 hPa) and by a local maximum in the stratosphere (100–1 hPa) which maximises during winter and spring. In the Southern hemisphere, the impact of the Antarctic ozone hole is clearly visibly as a nearly complete loss of ozone in the lowermost stratosphere (below ≈ 30 hPa) in early spring. These main structures are well represented by all models. However, the mesospheric winter-time maximum is overestimated by the high-top models (3dCTM and KASIMA) above ≈ 0.1 hPa by up to 2 ppm (nearly a factor of 2), while it is underestimated by EMAC by up to 1.5 ppm. In the lower mesosphere and at the stratopause (2–0.1 hPa), all three models underestimate ozone by 0.5 ppm to more than 2 ppm. This underestimation is largest (more than 2 ppm) in EMAC in the Northern hemisphere, where it also displays an annual variation with largest values in spring. In the region of the stratospheric maximum (100-2 hPa), 3dCTM overestimates ozone by 0.5–2 ppm in the

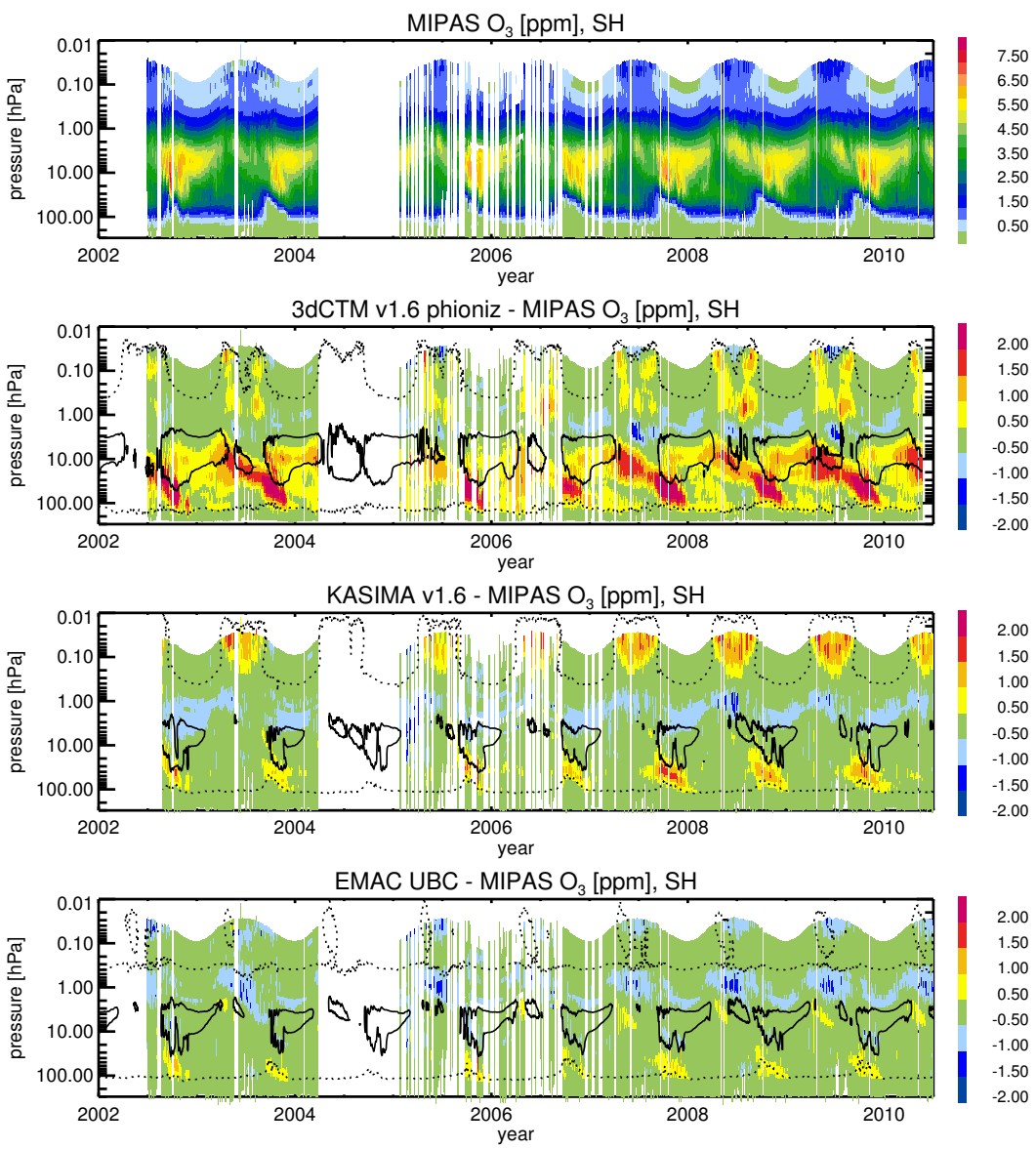

**Figure 8.** Comparison of ozone time-series at high Southern (70–90°) latitudes from MIPAS observations with model results including full particle forcing (3dCTM v1.6 phioniz, KASIMA v1.6, EMAC UBC). From top to bottom: MIPAS ozone mixing ratios (ppm), difference of 3dCTM, KASIMA and EMAC mixing ratios to MIPAS mixing ratios (ppm). Solid and dotted black lines are the 1 ppm and 0.5 ppm contours of the model fields.

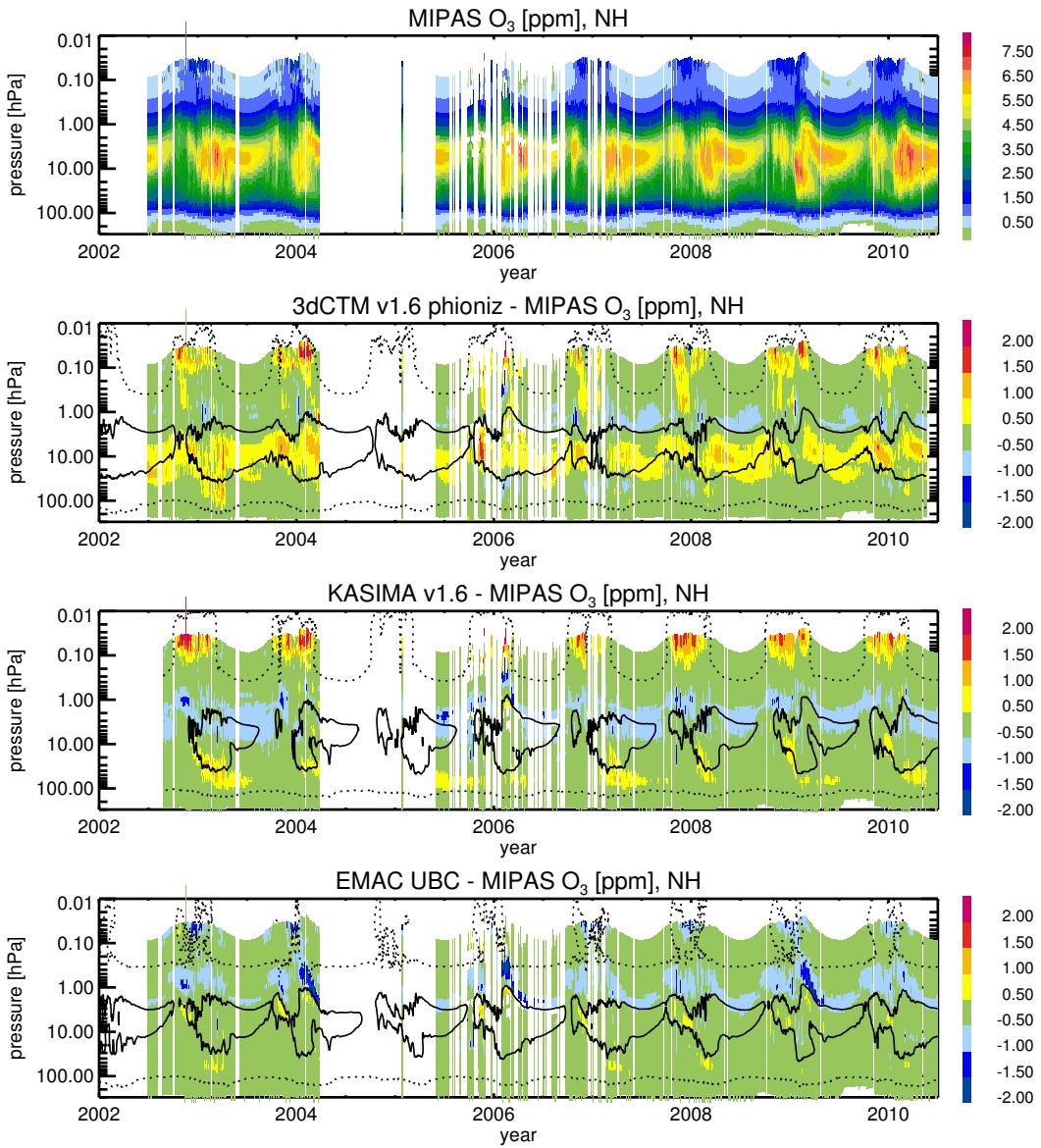

**Figure 9.** Same as Fig. 8, but for high Northern (70–90°) latitudes.

Southern hemisphere, by 0.5–1.5 ppm in the Northern hemisphere. KASIMA and EMAC generally reproduce the values of the stratospheric maximum well, with differences less than $\pm 1$ ppm. The strong overestimation of ozone in this region by 3dCTM might be due to the underestimation of $NO_y$ discussed in the previous chapter. Below the main amount of the stratospheric ozone layer (100-10 hPa during late winter and spring) ozone is overestimated by all models in the Southern hemisphere, possibly due to underestimation of the ozone loss in the Antarctic ozone hole region. This is most pronounced in 3dCTM (more than 2 ppm), with much lower values of 0.5–1 ppm in KASIMA and EMAC, and much less pronounced in the Northern hemisphere. In summary, all models display systematic differences (biases) in ozone compared to observations. However, it is unlikely that these biases are related to the treatment of energetic particles in the models, as the impact of energetic particle precipitation is usually – with the exception of strong solar proton events – masked by the much larger dynamical variability of ozone. In the next subsection, we will investigate this interannual variation of ozone, and how it relates to energetic particle precipitation; in the following subsection Sec. 4.3, the energetic particle impact is extracted from the model results by comparing the model runs with full particle forcing with the Base model scenarios.

## 4.2 Comparison of modeled and observed ozone anomalies

Due to the large dynamical variability of ozone particularly in the stratosphere, an impact of energetic particle precipitation on ozone can be difficult to derive from observations. Previous studies based on observations have compared observations of ozone in situations with and without elevated amounts of $NO_y$ on the same day and in the same latitude range (Natarajan et al., 2004), the evolution of ozone in years with high energetic particle fluxes compared to years with low particle fluxes (Randall et al., 2005), and the composite difference of years with high minus years with low particle fluxes (Fytterer et al., 2015a; Damiani et al., 2016). All methods have yielded lower values of ozone in the upper stratosphere presumably related to the energetic particle precipitation; the composite method also shows negative ozone anomalies proceeding downwards from the stratopause to the mid-stratosphere during polar winter. In the following, we will analyze ozone anomalies for the time-period 2002–2010 from MIPAS observations to investigate the interannual variation of ozone. Anomalies are calculated as follows. First, a climatology is built for every day of the year as the mean of the years 2006–2009, i.e., the January 1st data of all years are averaged to give the climatological value for January 1st, and so on. The period 2006–2009 was chosen for two reasons. MIPAS data are available nearly continuously during this period. Also, during this time geomagnetic activity as a proxy for particle forcing was low, so the ensuing climatology can be used as a reference for low particle forcing. Anomalies are calculated by subtracting the value of this climatology for every day of the year. The resulting percentage anomalies are shown in figures Fig. 10 and Fig. 11 for high Southern respectively Northern latitudes (70–90°S/N).

The strongest anomalies, of more than $\pm 30\%$, are observed in the time-period mid-2002 to early 2004 and in 2005 in both hemispheres, while in the period 2006-2010, anomalies are mostly in the range $\pm 20\%$ (SH)/(-30,+20) % (NH) above 10 hPa. Anomalies in the upper mesosphere are characterized by a strong annual variation with a change in sign from summer to winter; highest negative anomalies of more than -50 % are reached during Southern hemisphere winter 2003 and Northern hemisphere winter 2003–2004 in this altitude region. In the mid-mesosphere to mid-stratosphere (0.2–10 hPa), anomalies are characterized by subsequent positive and negative downwelling signals related to changes in the speed of the downward-

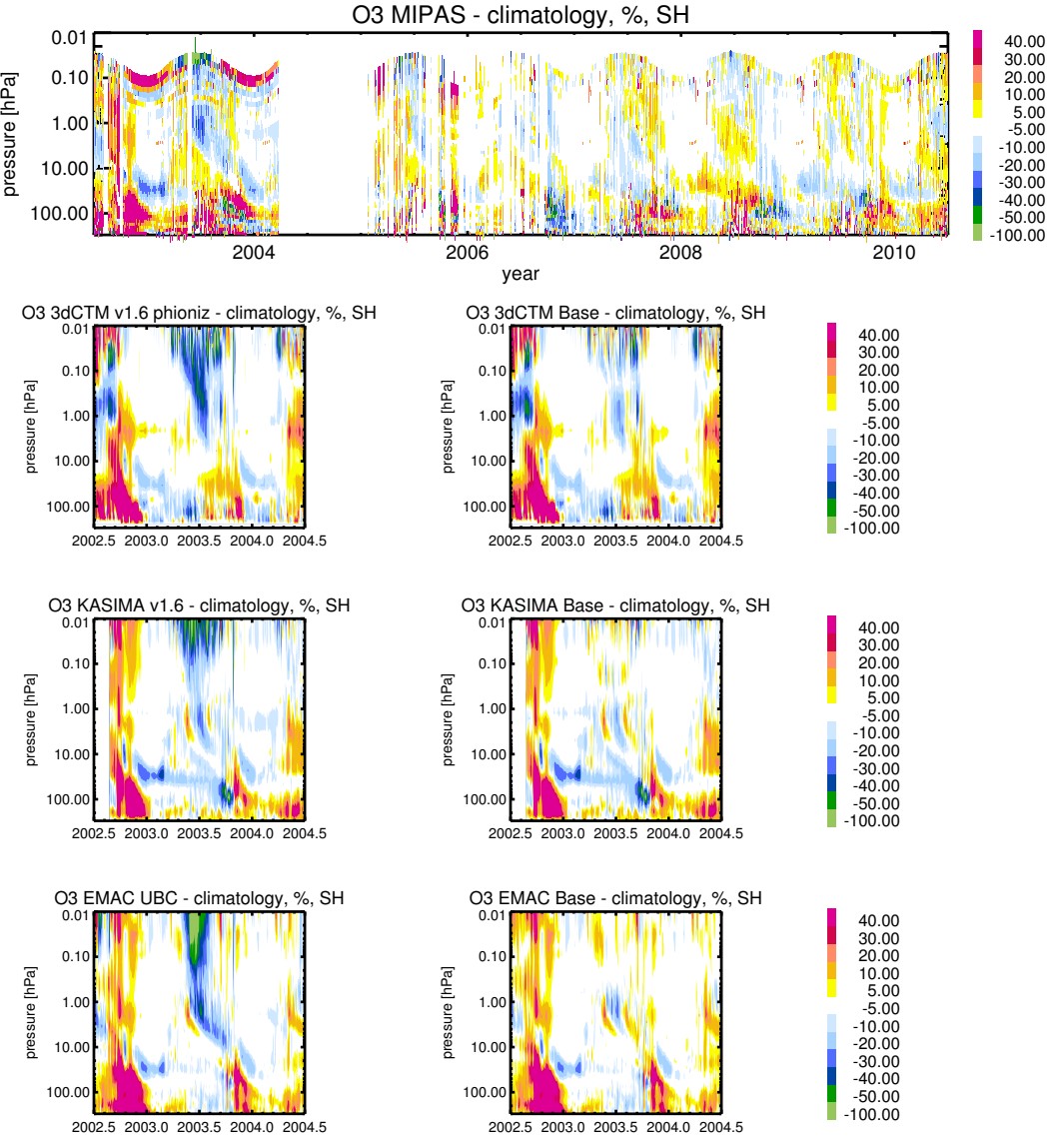

**Figure 10.** Upper panel: relative anomalies of MIPAS $O_3$ compared to a daily climatology derived from the 2006–2009 mean (%, see text) for high Southern latitudes (70–90°S) for the period 2002–mid 2010. Lower panels: relative anomalies for the model results, calculated in the same way for mid-2002 to mid-2004. Left: model runs including particle impact; right: model runs without particle impacts. From top to bottom: 3dCTM, KASIMA and EMAC.

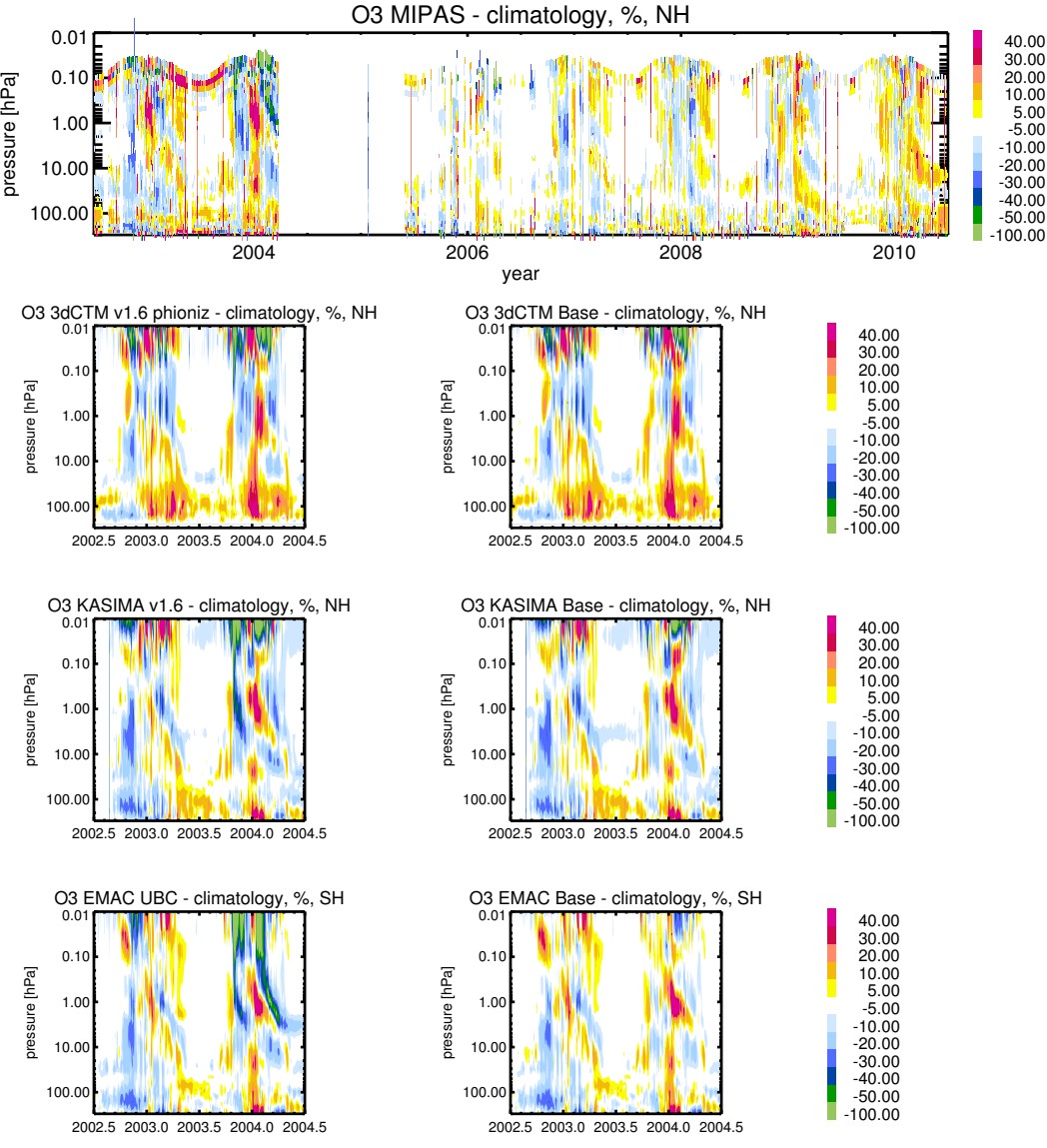

**Figure 11.** Upper panel: relative anomalies of MIPAS $O_3$ compared to a daily climatology derived from the 2006–2009 mean (%, see text) for high Southern latitudes (70–90°N) for the period 2002–mid 2010. Lower panels: relative anomalies for the model results, calculated in the same way for mid-2002 to mid-2004. Left: model runs including particle impact; right: model runs without particle impacts. From top to bottom: 3dCTM, KASIMA and EMAC.

poleward motion during polar winter. Below 10 hPa, very strong anomalies of more than $\pm$ 40 % occur mainly during winter and spring, particulary in the Southern hemisphere, and possibly related to interannual variations in the Antarctic ozone hole. A strong positive anomaly is observed, e.g., in late 2002 in the Southern hemisphere below 20 hPa which likely is related to the ozone-hole split during this Antarctic winter (B-M. Sinnhuber et al., 2002). A clear negative downwelling signal of

5-30 % is observed in Antarctic winter 2003 from the lower mesosphere ($\approx$ 0.3 hPa) in mid-winter to the lower stratosphere (below 10 hPa) in spring, preceded by a weaker (5-10 %) positive signal. The structure and strengths of this signal is similar to the downwelling anomalies derived from global satellite observations for the Southern hemisphere (Fytterer et al., 2015a; Damiani et al., 2016) when comparing composites of years with high minus low geomagnetic activity, and are interpreted as particle impacts there. In the Northern hemisphere, a similar negative downwelling signal starts in the lower mesosphere in

early winter 2003–2004, but is interrupted by a strong (> 40 %) positive anomaly in late 2003 and early 2004 probably related to the onset of the sudden stratospheric warming. In late winter and spring 2004, after the sudden stratospheric warming, an even stronger negative anomaly of more than -50 % descends down from the upper mesosphere (0.02 hPa) to the stratopause (1 hPa) in spring. However, from the observations it is not clear whether the strong anomalies during and after the warming are related to dynamical changes related to the warming, or due to the strong $NO_y$ signal observed after the warming. This will be

investigated in the following by comparing anomalies obtained in the same way from model results with and without particle forcing.

For the model data, anomalies are obtained in the same way as for the observations. First, a daily climatology is derived for the years 2006–2009 for the model runs with full particle forcing (3dCTM v1.6 phioniz, KASIMA v1.6, and EMAC UBC) and for the Base model runs. Percentage anomalies are shown for the period mid-2002 to mid-2004 in the lower left panels of

figures Fig. 10 and Fig. 11 for the model runs including full particle forcing, in the lower right panels for the Base model runs.

Model runs including energetic particle precipitation generally show a very consistent pattern of anomalies compared to the observations. In particular, all three models show very large negative ozone anomalies in the upper mesosphere ($\geq$ 0.1 hPa) in Southern hemisphere winter 2003, with values ranging from 20–40 % (3dCTM) to more than 40 % (KASIMA, EMAC) over a period of several weeks. Equally, all three models show very large positive ($\geq$ 40%) anomalies in late Southern hemisphere

winter 2002 in the lowermost stratosphere (200-10 hPa). All three models show a negative anomaly proceeding downwards from the lower mesosphere (0.1 hPa) in mid-winter 2003 to the lower stratosphere ($\leq$ 10 hPa) in late spring 2003 in the Southern hemisphere, preceded by a smaller positive anomaly in the upper stratosphere (10–1 hP). However, the strength of this negative anomaly as well as the apparent speed of its downward motion vary from model to model, consistent with differences in the descent rate in the stratosphere already discussed in section Sec. 3.3 for $NO_y$. Transport across the stratopause is restricted in

3dCTM, leading to too high anomalies in the lower mesosphere (1–0.1hPa) and a too small anomaly in the stratosphere; the downwelling speed is well captured in KASIMA, though the amplitude of the anomaly is too small compared to observations; while the stratospheric anomaly is too large in EMAC. In the Northern hemisphere, again all three models show very large negative ozone anomalies of more than -50 % in the upper mesosphere ($\geq$ 0.1 hPa) in winter 2003–2004. The general structure of the anomalies from the lower mesosphere to lower stratosphere is also well captured by all three models, with a sequence

of positive anomalies in the early winter stratosphere ($\approx$10–1 hPa) followed by a negative signal moving down from the

mesosphere to the upper stratosphere, which is interrupted due to the sudden stratospheric warming in December 2004, by a very strong positive signal, followed again by a negative signal after the sudden stratospheric warming. While the strengths of the positive signals seems to be well captured by all three models, the strengths of the negative signal preceding the warming again varies from model to model, with values generally lower than observed in 3dCTM, higher than observed in KASIMA and EMAC. The strong negative signal after the warming in the mid-stratosphere to lower mesosphere is strongly underestimated by 3dCTM and KASIMA, but well captured by EMAC, in good agreement with results shown for the downwelling signal of $NO_y$ discussed in Sec. 3.3.

The Base model runs without energetic particle precipitation show a very similar pattern of anomalies as the model runs with full particle forcing discussed in the previous paragraph, indicating that most of the anomalies observed are not due to chemical changes due to the particle forcing, but due to changes in dynamics (temperature and large-scale transport) from year to year. In particular, the anomalies below 10 hPa are nearly identical in model runs with and without particle forcing, indicating that all anomalies below this pressure level are due to dynamical changes. This includes a negative signal which moves down from about 10 hPa in late Southern hemisphere winter 2003; though this appears to be connected to the negative anomaly moving down from the mesosphere to the mid-stratosphere throughout the winter, it is apparently not due to chemical changes due to the energetic particle precipitation. As the specified temperatures and wind fields in all model experiments are based on the meteorological analyses in the stratosphere, it can not be ruled out that changes in dynamics from year to year are related to the particle precipitation, particularly in the years of strong particle forcing (2002–2004); however, it is not possible to assess this with these model experiments. A negative anomaly moving down from the mesosphere to the mid-stratosphere during Southern hemisphere winter 2003 and Northern hemisphere winter 2003-2004 is also observed in all three models in the model runs without particle forcing, indicating that these negative anomalies are partly due to changes in the downward transport speed or horizontal mixing. However, the strength of the negative anomalies is larger in the model runs including particle precipitation, indicating that energetic particle precipitation also contributes to these anomalies. In particular in Southern hemisphere winter 2003 and in the Northern hemisphere winter 2003–2004 after the sudden stratospheric warming, negative anomalies are much larger in the model runs with particle forcing than in the respective Base model runs, indicating that during these periods the particle impact plays a significant role. In most other winters, the differences between ozone anomalies in the model runs with and without particle forcings are much smaller (not shown here). The large negative anomalies in the upper mesosphere in Southern hemisphere winter 2003 and Northern hemisphere winter 2003–2004 are not observed in the model runs without particle forcing, so they are probably due solely to particle forcing. Because of the apparent quite strong year to year variations in ozone due to changes in the vertical and poleward transport during polar winters even in the Southern hemisphere seen here as downwelling positive/negative anomalies, it is very difficult to extract a signal of particle precipitation from stratospheric and mesospheric ozone by comparing years with and without particle precipitation; as the comparison of the analysis of model runs without particle forcing shows, downwelling negative anomalies can be produced due to dynamical reasons, and falsely attributed to particle forcing. In the following Section we investigate the impact of energetic particle precipitation on ozone in the stratosphere by comparing model runs with and model runs without particle forcing; as the model runs use specified

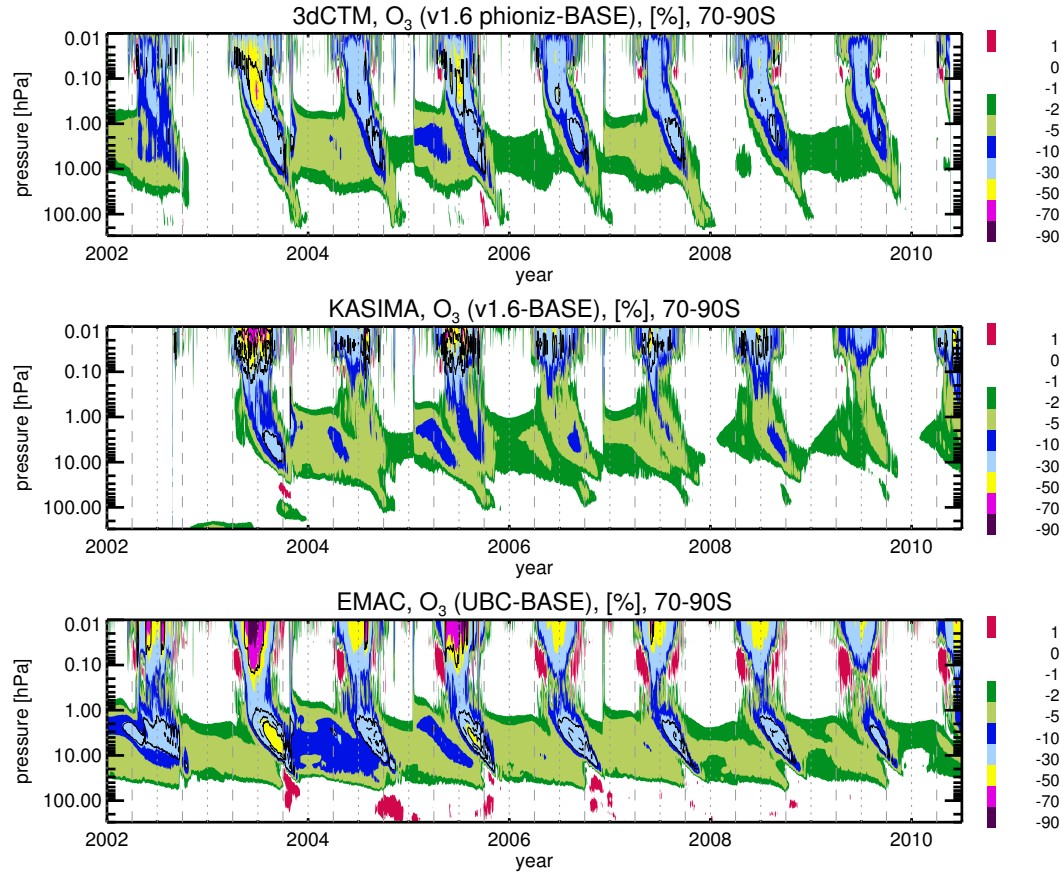

**Figure 12.** Relative Ozone anomalies due to energetic particle precipitation at high Southern latitudes (70–90°S), calculated as the difference of model runs with to without particle impact (%). Top: 3dCTM (v1.6 phioniz-Base), middle: KASIMA (v1.6 - Base), third: EMAC (UBC - Base). The gray shaded area in the middle figure denotes the time-period before the start of the KASIMA model runs. Dashed and dotted vertical gray lines mark zero, one-quarter, one-half, and three-quarters of each year. The thin black contours refer to -1.5 (solid), -1 (dashed) and -0.5 (solid) ppm.

dynamics, the dynamical variability of ozone from year to year is the same in both model experiments, and the difference between the model experiments highlights the particle impact only.

## 4.3 Modeled Ozone anomalies due to particle precipitation

Ozone anomalies due to energetic particle precipitation only are derived from the model results as the difference of model runs with (3dCTM v1.6 phioniz, KASIMA v1.6, EMAC UBC) to the respective Base model run without particle impacts. Percentage differences are shown for all three models in high Southern latitudes (70–90°S, Fig. 12) and in high Northern latitudes (70–90°N, Fig.13).

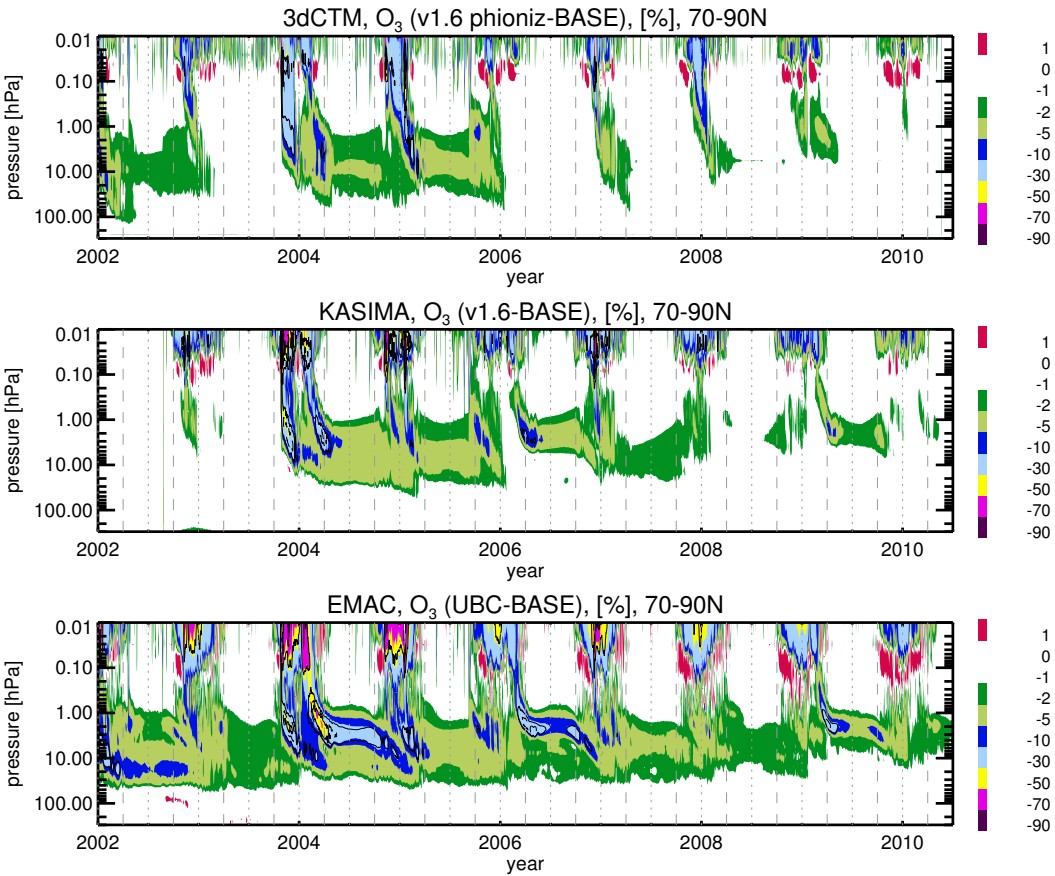

**Figure 13.** Relative Ozone anomalies due to energetic particle precipitation at high Northern latitudes (70–90°N), calculated as the difference of model runs with to without particle impact (%). Top: 3dCTM (v1.6 phioniz-Base), middle: KASIMA (v1.6 - Base), third: EMAC (UBC - Base). The gray shaded area in the middle figure denotes the time-period before the start of the KASIMA model runs. Dashed and dotted vertical gray lines mark zero, one-quarter, one-half, and three-quarters of each year. The thin black contours refer to -1.5 (solid), -1 (dashed) and -0.5 (solid) ppm.

In high Southern latitudes, all three models show a consistent behaviour with a distinct annual/vertical variation in the ozone loss, as well as a few sporadic events of short-lived ozone loss:

- In the upper mesosphere above 0.1 hPa, all three models predict strong ozone losses during polar winter. The vertical structure and strength of these winter-time mesospheric ozone losses vary from year to year, and between models from 10–30 % in winter 2009 (KASIMA, 3dCTM) to 70–90 %/50–70 % in winter 2003 (EMAC/KASIMA). In KASIMA and EMAC, mesospheric ozone loss is constant or increases with altitude above 0.1 hPa, while in 3dCTM, the ozone loss decreases again above $\approx 0.03$ hPa, maximizing around 0.1 hPa.

- The negative ozone anomaly simulated in every winter moves downward from the mesospheric ozone loss region in early winter, to the mid-stratosphere ($\approx 10$ hPa) in late winter. Again, values vary from winter to winter and between models, with highest values of 30–50 % reached in the upper stratosphere (2–6 hPa) in the solar maximum winter 2003 (EMAC), lowest values of 2–5 % reached in the solar minimum winter 2009 (KASIMA). The temporal structure and vertical extent of these downwelling negative anomalies are in good agreement with the observations of negative stratospheric ozone anomalies relative to enhanced geomagnetic activity, see Fytterer et al. (2015a). However, absolute values can not be compared directly because Fytterer et al. (2015a) investigates the interannual variation, while here, the difference between ozone with and without particle forcing is investigated for the same year.

- Absolute differences in the stratospheric winter-time ozone anomalies are largest in EMAC (more than 0.5 ppm in all winters, more than 1 ppm in all winters but 2009, more than 1.5 ppm in winters 2003 and 2005 between 10–1 hPa), lower in 3dCTM (0.5–1 ppm in all winters but 2009), lowest in KASIMA (more than 0.5 ppm only in winter 2003). Values for winter 2003 are in good agreement with a previous model study incorporating MIPAS $NO_x$ in the lower mesosphere and showing 1–1.5 ppm ozone loss between 30–40 km compared to a model run without excess $NO_2$ (Reddmann et al., 2010).

- Strong solar proton events in October 2003, January 2005 and December 2006 lead to an instantaneous loss of ozone from the upper stratosphere to the mesopause (10–0.01 hPa). In the mesosphere, this impact is shortlived and restricted to a few days only, while in the stratosphere below 1 hPa, ozone loss continues throughout the summer.

- After winters with a strong stratospheric ozone loss signal, ozone loss of 2–5 % continues through Antarctic spring and summer in the mid-and upper stratosphere (20–1 hPa) until the next winter. In a few summers, this mid-stratospheric summer ozone loss can reach values of 5–10 %, in particular in early 2004 and early 2005. In these summers, the continuing ozone loss from the indirect effect seems to be strengthened by strong solar proton events occuring in early spring (October 2003) respectively during summer (January 2005).

- Small regions of a positive ozone change are observed in the lower mesosphere in early and late winter below the regions of strong mesospheric winter-time ozone loss (3dCTM, EMAC); these are likely due to self-healing, i.e., stronger ozone formation below regions of ozone loss because of the stronger UV radiation.

Mesospheric ozone loss has been observed related to strong energetic electron precipitation events (Andersson et al., 2014b) and also related to the 27-day cycle of the geomagnetic activity (Fytterer et al., 2015b). It is also predicted by model studies (Semeniuk et al., 2011; Fytterer et al., 2016; Arsenovich et al., 2016), and is likely related to the increase in $HO_x$ during electron precipitation events. Is is restricted mainly to polar winter because during summer, the background in $HO_x$ is higher, therefore the relative increase in $HO_x$ due to the electron precipitation is rather smaller (Fytterer et al., 2015b, 2016). The values simulated here are in range of the observations, which show ozone losses of about 10 % averaged over one winter, up to 90 % in individual strong events (Andersson et al., 2014b). The mesospheric ozone loss also agrees well with model results by the CMAM model driven with prescribed auroral and medium-electron ionization for the period 1979–2006 (Semeniuk et al., 2011). They show a multi-annual mean of 10–30 % (Northern hemisphere) respectively 30–80 % (Southern hemisphere) of ozone loss compared to model runs without particle impacts in mid-winter (DJF respectively JJA) in the upper mesosphere above $\approx$65 km (about 0.1 hPa). Particle-induced ozone loss in the mesosphere is mainly due to catalytic cycles involving $HO_x$, which is released from positive water cluster ions formed by incorporating water vapor, thus depends on the availability of water vapor (Solomon et al., 1981; Sinnhuber et al., 2012). It has been shown recently that during polar winter, mesospheric ozone loss can also be initiated by $NO_x$, indirectly by changing the partitioning of $HO_x$ from $HO_2$ to OH (Verronen and Lehmann, 2015). Differences in the vertical structure of the ozone loss between KASIMA and EMAC might therefore denote different gradients in the mesospheric water vapor and $NO_y$ content in the models. However, the strong mesospheric ozone loss predicted by EMAC is more likely due to the implementation of the upper boundary condition: in every time-step, NO in EMAC is overwritten by the upper boundary $NO_y$. To balance $NO_x$, other $NO_x$ species like, e.g., $NO_2$ are set to zero. This leads to realistic values of $NO_y$ as shown in Sec. 3.4; however, in every chemistry time-step, the reactions $NO + O_3 \longrightarrow NO_2$ + O and $NO + HO_2 \longrightarrow NO_2 + OH$ are also processed, changing the partitioning of $HO_x$ and destroying larger amounts of ozone than in the Base run.

Ozone loss in the mid-stratosphere is mainly due to catalytic cycles involving $NO_x$, and the stratospheric ozone loss predicted by the models can be directly related to the EPP-$NO_y$ brought into the stratosphere by the indirect effect and by large solar proton events. The winter-time ozone loss seems to be mainly due to the indirect effect, and this continues well into summer in agreement with the long-term accumulation of EPP-$NO_y$ discussed in the previous chapter. Strong summer-time ozone loss, e.g., in summer 2003–2004 and 2004–2005, nevertheless seems to be due mainly to strong solar proton events.

In the Northern hemisphere, mesospheric ozone loss throughout the winter is predicted by all models similar to the Southern hemisphere. Solar proton events in October 2003, January 2005 and September 2005 lead to instantaneous ozone loss from the mid-stratosphere to the upper mesosphere in all models. However, the annual variation of the stratospheric ozone loss due to the indirect effect looks distinctly different. In the Southern hemisphere, a continuous downwelling negative anomaly reaching down to the mid-stratosphere (10 hPa) is observed in every winter. In the Northern hemisphere, this is not the case. In the solar minimum winters (2006–2007 to 2009–2010), stratospheric ozone loss is significantly lower than in the Southern hemisphere winters, and the signal does not reach down to 10 hPa in most of these winters. In the solar maximum winters, strong stratospheric ozone loss of more than 50 % (late winter 2003-2004 in EMAC), 30–50 % (winter 2003-2004 in KASIMA and early winter 2003-2004 in EMAC) or 10–30 % (winters 2003-2004 in 3dCTM, 2004-2005 in 3dCTM and EMAC) is

predicted, and this ozone loss continues well into summer. However, the structure of the downwelling signals in these winters is distinctly different to the structure in Southern hemisphere winters with two distinct peaks of ozone loss, one in early winter, one in late winter. The second peak in late winter 2003–2004 is due to the strong downwelling of $NO_y$ from the mesosphere after the strong sudden stratospheric warming. It is stronger in EMAC than in KASIMA, and weakest in 3dCTM, following the

differences in $NO_y$ in the models. In EMAC, downwelling ozone losses of 10–30% are also observed outside solar maximum related to the sudden stratospheric warmings in early 2006 and early 2009; these are weaker in KASIMA, and not observed in 3dCTM. After the sudden stratospheric warmings in early 2004, early 2006, and early 2009, stratospheric ozone loss of 5–30 % continues throughout Arctic spring and summer in EMAC. However, in other summers, continuing stratospheric ozone loss from the EPP indirect effect is lower than in the Southern hemisphere, less than 1 % in some summers (2006-2010 in

3dCTM, 2008 and 2010 in KASIMA). Absolute differences during winter again range from less than 0.5 ppm (all but 2003–2004 and 2004–2005 in 3dCTM, all but 2003–2004 and 2005–2006 in KASIMA, 2002–2003, 2007–2008, and 2009–2010 in EMAC) to more than 1 ppm (KASIMA) respectively more than 1.5 ppm (EMAC) after the sudden stratospheric warming in winter 2003–2004. These values are in good agreement with observations of ozone variations of 1–2 ppm in longitudes with enhanced $NO_y$ compared to airparcels without enhanced $NO_y$ in early April 2004 at 2 hPa (Natarajan et al., 2004) as well

as with observed differences of about 1 ppm of ozone mixing ratios in March and April 2004 compared to previous winters at $\approx 40$ km (Randall et al., 2005). The structure of the downwelling signal appears to be consistent with a previous model study incorporating MIPAS $NO_x$ in the lower mesosphere and showing 2–3 ppm ozone loss between 35–40 km compared to a model run without excess $NO_2$ in Northern hemisphere winter 2003–2004 (Reddmann et al., 2010). These results agree well with EMAC results which also reach more than 2 ppm, but are higher than KASIMA or 3dCTM results. On average, ozone

loss in the mid-to upper stratosphere during mid-winter agrees well with results from the CMAM model (5–30 % during JJA in 30-40 km in a 1979-2006 multi-year average in the Southern hemisphere, 0.5–5 % in the Northern hemisphere).

To summarize, stratospheric ozone loss due to energetic particle precipitation is predicted by all three models in most winters in both hemispheres, but the vertical range, temporal structure, and strength of the ozone loss varies from year to year, between models, and between hemispheres. Stratospheric ozone loss often continues into polar summer. In the Southern hemisphere,

strong summer-time ozone losses are related mainly to strong solar proton events, while in the Northern hemisphere they are related mainly to strong sudden stratospheric warmings. In the Northern hemisphere, wintertime stratospheric ozone loss seems to be dominated by sudden stratospheric warmings as well, and is small in winters without strong warmings. In contrast, in the Southern hemisphere, the wintertime stratospheric ozone loss is dominated by continuous downwelling of $NO_y$ from the mesosphere, and is predicted by all models to occur in every winter.

**4.4   Changes in net radiative heating**

In the following, changes in the net radiative heating due to changes in ozone are derived from the modeled ozone changes discussed in the previous paragraph. All three models calculate radiative heating and cooling rates, but use different spectral resolutions and parameterizations. To obtain results for all three models independent of differences in the parameterizations, the shortwave heating rate in the Hartley bands of ozone ($\lambda \leq 320\ nm$) and the longwave radiative cooling in the $\nu = 001 \rightarrow$

$\nu = 000$ transition of ozone at 1042 cm$^{-1}$ (9.6 $\mu$m) are estimated using the daily zonally averaged ozone and temperature fields of the respective model as follows.

In a first step, the change in radiative flux in each model box is calculated depending on the temperature in the box center, and the column density of ozone between the upper and lower box boundaries for each model scenario.

For the shortwave radiation in the Hartley bands of ozone, the change in radiative flux is derived from the amount of downwelling solar radiation absorbed in the box:

$$\Delta F_{Hartley} = J_{O_3} O_{3,col} \Delta E, \tag{1}$$

where $J_{O_3}$ is the daily mean Hartley band photolysis rate, $O_{3,col}$ is the column density of ozone between upper and lower box boundaries, $\Delta E$ is the energy transferred into heat, and $\Delta F$ is the change in radiative flux in $Wm^{-2}$. The amount of energy transferred into heat is estimated as the mean energy of a photon in the Hartley bands, at 260 nm ($7.64 \times 10^{-19} J$)[3]

The daily mean photolysis rate is calculated using the photolysis scheme of 3dCTM for the respective latitude and day of year using fixed ozone, density, and temperature profiles by calculating photolysis rates every five minutes and averaging over a full day. The Huggins and Chappuis bands have not been taken into account here because they contribute to solar heating only in the lower stratosphere and below (Brasseur and Solomon, 2005).

For the longwave radiation, in a first step the upwelling flux is calculated from 200 hPa up to 0.01 hPa, and the downwelling flux from 0.01 hPa down to 200 hPa. The limits are chosen to make results from all three models comparable: from the upper limit of EMAC down to the lower limit of 3dCTM. As we are interested only in the mid-to upper stratosphere, the emission of thermal radiation at the surface is not considered, and the upward and downward fluxes can be calculated as

$$\Delta F_{001,out}(\nu) = (\Delta F_{001,in}(\nu) + \pi B(\nu,T)\sigma_{001}(\nu)O_{3,col})e^{-\sigma_{001}(\nu)\frac{O_{3,col}}{cos(53°)}} \tag{2}$$

where $\Delta F_{001,out}(\nu)$ is the flux out of the box at wavenumber $\nu$ in $W/m^2 cm^{-1}$, $\Delta F_{001,in}$ is the flux into the box, $B(\nu,T)$ is the Planck function at wavenumber $\nu$ and temperature T, and $\sigma_{001}(\nu)$ is the absorption cross section at wavenumber $\nu$, considering only the $v = 001 \rightarrow v = 000$ transition. Line-by-line absorption cross sections have been taken from the HITRAN database (*hitran.org*, February 2017, Rothman et al. (2013)), and binned into 1.5 cm$^{-1}$ intervals. A mean airmass factor of $\frac{1}{cos(53°)}$ is used as an approximation of the integral over the whole sphere. The radiative flux of the whole band is obtained by integrating over wavenumber from 950–1098.5 cm$^{-1}$. Temperatures of the Base model scenario are used. The change of flux in each box is then obtained by the differences of ingoing and outgoing upward and downward flux. It should be stressed that for a more exact calculation of the longwave cooling due to ozone, a line-by-line calculation of the $v = 100 \rightarrow v = 000$, $v = 010 \rightarrow v = 000$, and $v = 100 \rightarrow v = 000$ transitions should be carried out; however, results obtained here agree mostly to about $\pm 20$ % with calculations performed with the GRANADA non-LTE model using climatological profiles as described in Funke et al. (2012) (not shown here).

---

[3]It should be noted that in the mesosphere and above, this has to be balanced by the energy needed to dissociate ozone. However, in the stratosphere, this energy is immediately released again as chemical energy.

Heating or cooling rates are obtained from the fluxes as

$$\frac{\partial T}{\partial t} = \frac{g \Delta F}{\Delta p c_p} \tag{3}$$

where $c_p$ is the specific heat capacity of air, g the acceleration of gravity taken to be constant as 9,81 m/s, $\Delta p$ the pressure width of the box, and $\frac{\partial T}{\partial t}$ is the heating rate in $K/s$. It should be pointed out that above about 60 km altitude, non-LTE effects as well as the diurnal variability of ozone become increasingly important both for the shortwave heating and for the longwave cooling terms. These are not considered here, and only results below $\approx 0.1$ hPa are considered in the following.

The changes in net radiative heating and cooling rates due to particle precipitation are calculated as the difference between model runs with to without particle impacts, and the net heating rate change is the sum of changes in heating and cooling rates.

The net changes in heating rates due to the particle-induced ozone loss are shown for high Southern (70–90°S) and high Northern (70–90°N) latitudes in Figs. 14 and 15. For all three models and in both hemispheres, changes in the net heating are confined to the upper stratosphere and above (above 20 hPa), and peak around the stratopause (around 1 hPa). Net heating rate changes show a clear annual variation with positive values (a net heating) during winter, and negative values (a net cooling) during spring. The net heating during winter is due to a decrease in radiative (longwave) cooling due to the ozone reduction, while the net cooling during spring is due mainly due to a decrease in radiative (shortwave) heating, which is balanced to some extent by the decrease in radiative (longwave) cooling. In some years, the net cooling of the spring upper stratosphere continues well into summer. In the Southern hemisphere, these are years with strong solar proton events in spring or summer (2004, 2005, 2007); in the Northern hemisphere, these are mainly years with strong sudden stratospheric warmings (2004, 2006, 2009), but also one year with with a solar proton event in late winter (2005). The strongest and longest-lasting impacts of more than 0.6 K/day (EMAC) respectively more than 0.1 K/day (KASIMA) for several month is predicted for the sudden stratospheric warmings in Northern hemisphere winters 2003–2004, 2005–2006, and 2008–2009. The solar proton event of October 2003 has an impact comparable with the sudden stratospheric warmings as predicted by EMAC on net heating rates of more than 0.5 K/day, but lasting a few days only, and predicted by all three models. Values of 0.1–0.2 K/day are then predicted to continue throughout the summer. In the lowermost panels of Figs. 14 and 15, the daily averaged sum of all shortwave and longwave contributions to radiative heating and cooling (called net radiative heating rate in the following) are shown for comparison for the EMAC Base scenario. The strongest changes of the net radiative heating rate due to energetic particle precipitation during mid-winter occur in the upper stratosphere and lower mesosphere (10–0.1 hPa), at the lower edge of a region of strong radiative cooling in the lower mesosphere. The net radiative heating rates increase from 1 K/day at 10 hPa to more than 10 K/day at 1 hPa. Changes due to particle precipitation are in the order of magnitude of 0.1 K/day in this altitude range, reaching more than 0.5 K/day in Southern hemisphere winter 2003 (EMAC). This amounts to relative changes of 1–10 % in most winters, up to 50 % in winter 2003 (EMAC). In contrast, negative changes to the net radiative heating during spring after strong solar proton events or sudden stratospheric warmings occur in a region of low net radiative heating rates ranging from less than 0.5 K/day to 1–2.5 K/day; changes after the strong solar proton event in October 2003 in the Southern hemisphere exceed 20 %, and changes after the strong sudden stratospheric warming in Northern hemisphere winter 2003–2004 and 2005–2006 approach 100 %. The continuing cooling of $\approx 0.1$ K/day in the upper stratosphere after the sudden stratospheric warmings throughout the

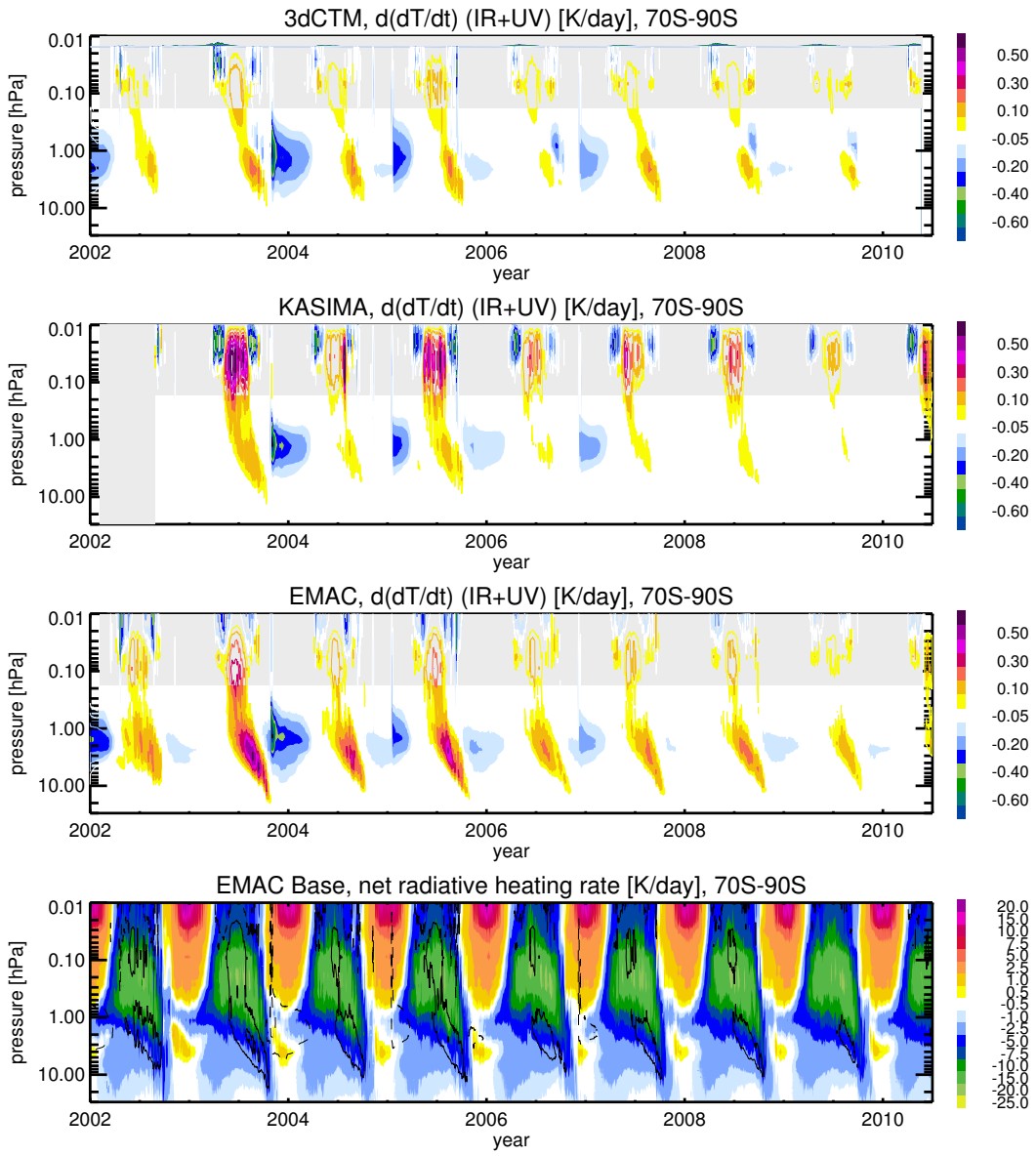

**Figure 14.** Changes in daily net radiative (short-wave and long-wave) heating rates (K/day) due to particle-induced ozone changes derived from all three models (top: 3dCTM, second from top: KASIMA, second from bottom: EMAC) at high Southern latitudes (70–90°S). Bottom: The net radiative heating rate (sum of the shortwave and longwave contributions) in the EMAC Base scenario (K/day). Black dashed lines are the -0.1 and -0.5 K/day contours of the net radiative heating rate change in EMAC, black solid lines are the +0.1 and +0.5 K/day contours.

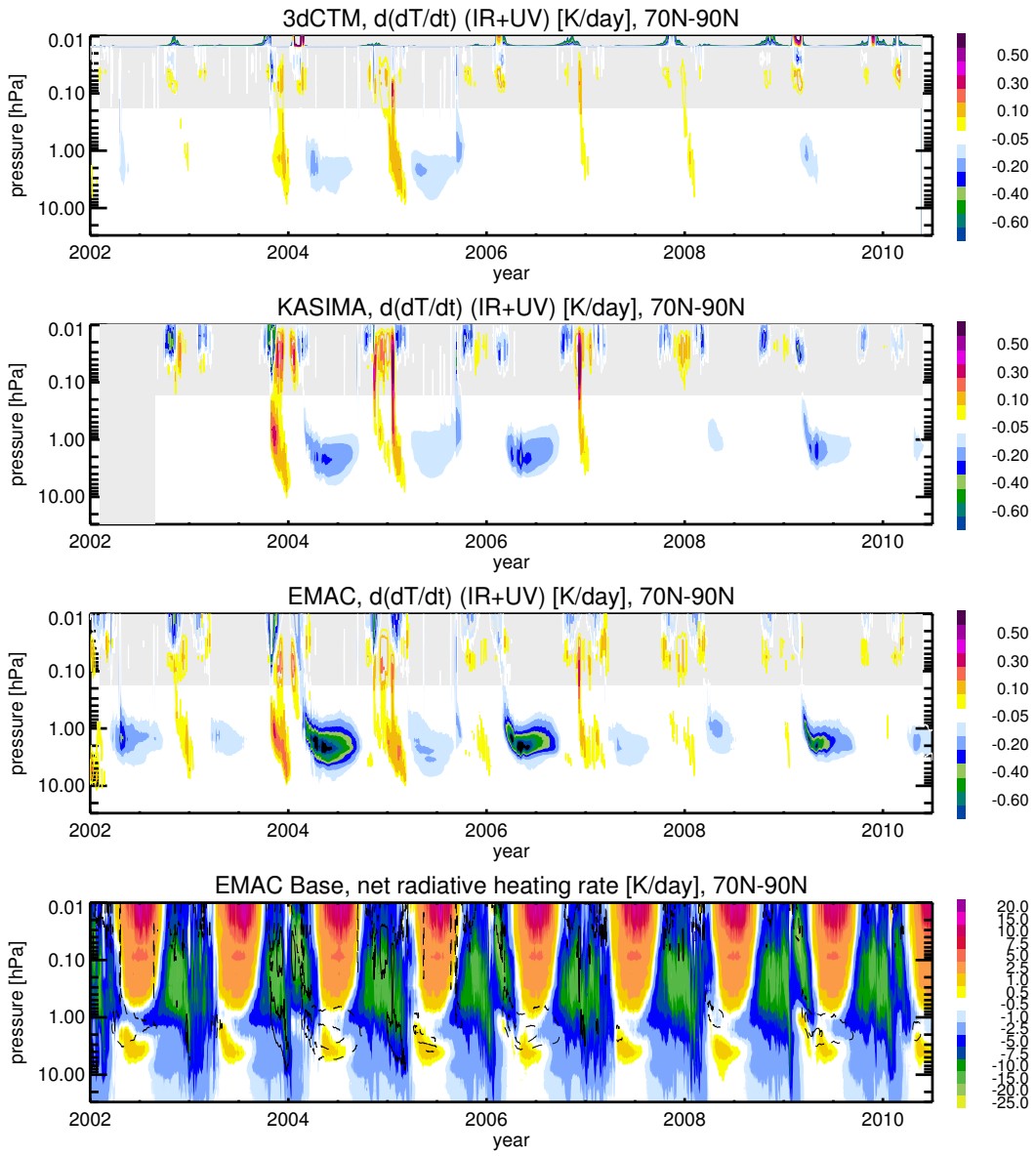

**Figure 15.** Changes in daily net radiative (short-wave and long-wave) heating rates (K/day) due to particle-induced ozone changes derived from all three models (top: 3dCTM, second from top: KASIMA, second from bottom: EMAC) at high Northern latitudes (70–90°N). Bottom: The net radiative heating rate (sum of the shortwave and longwave contributions) in the EMAC Base scenario (K/day). Black dashed lines are the -0.1 and -0.5 K/day contours of the net radiative heating rate change in EMAC, black solid lines are the +0.1 and +0.5 K/day contours.

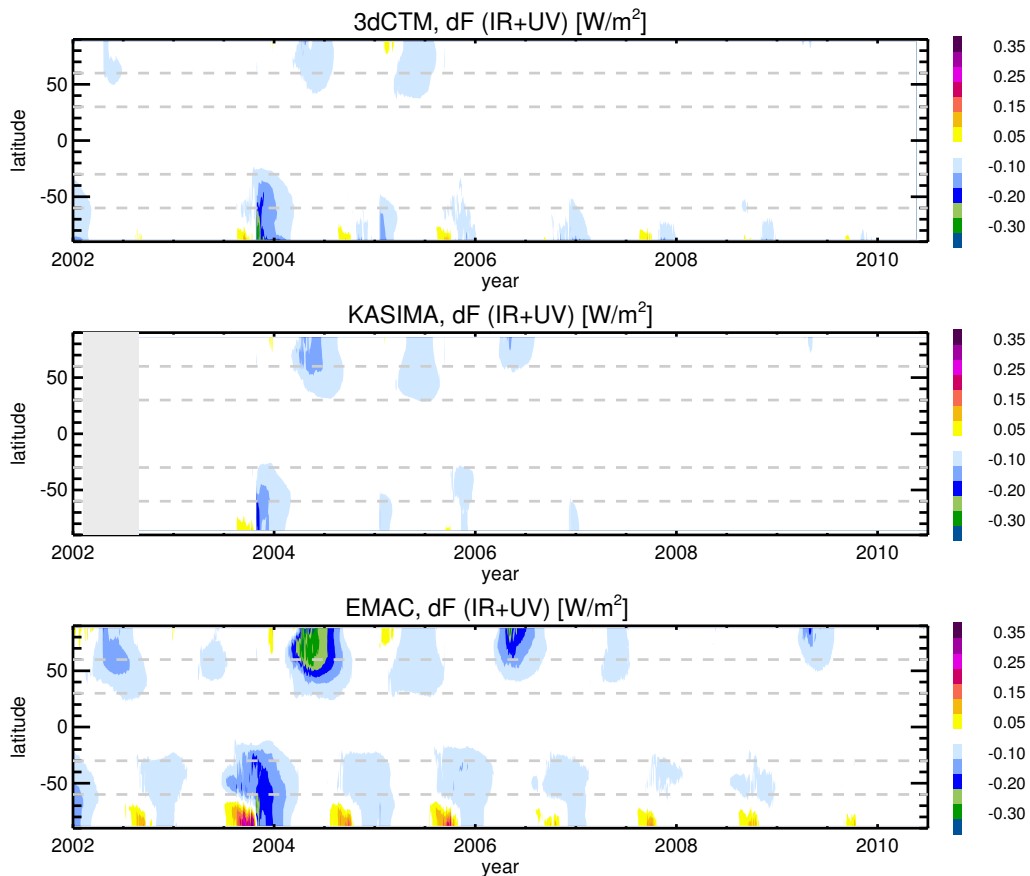

**Figure 16.** Daily changes in the net amount of energy absorbed or transmitted due to particle induced ozone destruction in the stratosphere (W/m$^2$), derived from the difference of model runs with to model runs without particle forcing, as a function of time and latitude.

summer are in the range of 4–10 % of the net radiative rate of 1–2.5 K/day at this time. During spring the upper stratosphere approaches radiative equilibrium, so small changes in the net radiation budget – as after sudden stratospheric warmings or wintertime solar proton events – could potentially have a large impact.

To investigate the temporal-spatial structure of the particle impact onto the net energy absorbed or emitted, the changes in radiative flux derived above for both the shortwave and longwave component are added up over the vertical region from 200 hPa to 0.2 hPa. The upper limit was chosen because above this region, the simplified approach used here is no longer valid, see above. The resulting changes in the net flux are shown in Fig. 16. Results confirm many of the findings already discussed for the heating rates: largest changes to the net energy absorbed are related to strong solar proton events in the Southern hemisphere, to sudden stratospheric warmings in the Northern hemisphere. Positive changes (net heating) are predicted for winter-time, but are restricted to high latitudes poleward of 60°. Negative changes (net cooling) are predicted mainly for spring and summer, and for latitudes equatorwards of 60°. The net cooling reaches well into midlatitudes (30°) in some years (2003 and 2006 in

the Southern hemisphere, 2004 and 2005 in the Northern hemisphere). This latitudinal extent of the particle impact well into midlatitudes (up to 30°) is consistent with results of EPP $NO_y$ derived from MIPAS observations as shown in Funke et al. (2014a) (their Fig. 12).

## 5   Conclusions

Analysis of results from three global chemistry-climate models driven by geomagnetic forcing and solar protons shows that solar proton events and the indirect effect due to downwelling of $NO_y$ presumably from the aurora contribute similar amounts of $NO_y$ to the stratosphere. Comparison to MIPAS $NO_y$ shows that on average, the indirect effect is captured well by the models in dynamically quiet winters, though the amount transported into the stratosphere depends on the treatment of gravity wave drag in the respective models. After sudden stratospheric warmings, the indirect effect is strongly underestimated by the two high-top models (3dCTM and KASIMA), while it is well represented by the medium-top model EMAC using the upper boundary conditions from Funke et al. (2016). Ozone loss in the mid-to upper stratosphere related to the indirect effect is predicted by all three models for all polar winters, varying from 10–50 % during solar maximum, to 2–10 % during solar minimum. Ozone loss maximizes in mid-winter to spring. The ozone losses lead to changes in the net radiative heating which change sign in late winter: a warming of the upper stratosphere at high latitudes dominates in mid-winters, while a cooling extending into midlatitudes dominates in late winter and spring. Analysis of several decades of re-analysis data show a warming of the mid-to late winter upper stratosphere related to high geomagnetic activity (Lu et al., 2008; Seppälä et al., 2013) which is also reproduced in model experiments using free-running chemistry-climate models (Semeniuk et al., 2011; Baumgaertner et al., 2011). Based on older model experiments by (Langematz et al., 2005), Baumgaertner et al. (2011) and Seppälä et al. (2013) argue that the warming in the upper stratosphere and lower mesosphere is consistent with a direct radiative impact, while a cooling of the middle and lower stratosphere observed at the same time, during mid-winter (DJF in the Northern hemisphere), is more likely the result of coupling between the vortex strength and wave propagation and reflection, an assumption strengthened by the apparent relation to the phase of the stratospheric quasi-biennial oscillation and the solar cycle (Lu et al., 2008; Seppälä et al., 2013). Our results are consistent with these earlier studies, and strengthen the assumption that the mid-winter warming is due at least partly to a direct radiative impact.

The indirect effect could contribute to the reformation of a strong and long-lasting vortex in late winter and spring after sudden stratospheric warmings in years with high geomagnetic activity, as e.g. in winter 2003–2004. Sudden stratospheric warmings in Northern hemisphere winter and spring as well as solar proton events occuring during spring or summer can lead to an ongoing cooling from spring to late summer, possibly pre-conditioning the stratosphere in autumn.

*Code and data availability.* The model output used here as well as idl routines used to calculate heating and cooling rates are available from M. Sinnhuber (*miriam.sinnhuber@kit.edu*) on request. AIMOS v1.6 ionization rates due to protons and electrons have been obtained from J. Wissing (*jawissin@uos.de*). Solar proton event dates have been taken from *http://umbra.nascom.nasa.gov/SEP/*, and

the proton ionization rates used in EMAC have been obtained from *http://solarisheppa.geomar.de/cmip6*. The sunspot number is from *www.ngdc.noaa.gov/stp/spidr.html*, the AE index from the Kyooto database at *wdc.kugi.kyoto-u.ac.jp*. MIPAS data processed at KIT are available at *imk-asf.kit.edu/english/308.php*.

*Competing interests.* Bernd Funke, Gabriele Stiller, and Thomas von Clarmann are Co-Editors of ACP, but are not involved in the editorial
5  work of this particular paper.

## 6 Acknowledgements

M. Sinnhuber gratefully acknowledges funding by the Helmholtz Association of German Research Centres (HGF), grant VH-NG-624. H. Nieder was funded by the project ROMIC-SOLIC (01LG1219C) funded by the German Ministry of Education and Research BMBF. The intercomparison was initialised at the *International Space Sciences Institute* in Bern in 2014, during
10  a meeting of the working group *Quantifying hemispheric differences in particle forcing effects on stratospheric ozone* led by D. Marsh. The authors acknowledge support by the state of Baden-Württemberg through bwHPC.

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
