# Peer review of "$NO_y$ production, ozone loss and changes in net radiative heating due to energetic particle precipitation in 2002-2010"

_Atmospheric Chemistry and Physics, 2017_

## Referee Comment (RC1) · Anonymous Referee #1 · 12 Jul 2017

Summary:

This paper uses three models nudged to reanalysis data to simulate electron and SPE propton effects on chemical composition of the middle atmosphere and the associated radiative impact. In particular, there is comparison of modeled and observed NOy as well as the impact of EPP on ozone. The authors show that the EPP indirect effect can account for some of the warming and cooling signal previously attributed to dynamics.

I recommend that the article be published subject to additional discussion of the issues raised below.

Major Comment:

On page 26 (around line 20) the authors speculate that application of the boundary condition in EMAC at 80 km may be resulting in excessive ozone loss. Semeniuk et al. (2011) using CMAM simulated large ozone losses around 50% at 80 km in the winter polar caps (70-90) South and North due to EPP HOx. Peak loss values were over 70% which agrees with the EMAC predicted losses averaged over the same period. CMAM did not use a boundary condition to drive EPP NOx but used the Jackman scheme for EPP NOx production and the Solomon scheme for EPP HOx production. So author's speculation about excessive O3 destruction based on excessive activity of the NO + O3 reaction does not apply to the CMAM and hence appears to be dubious. I suggest the authors either change the discussion of this aspect or conduct a more detailed chemistry analysis.

At the same time the ozone loss in the stratosphere aside from the peak years (2004-2005, 2006-2007) in the northern hemisphere is below 10% (closer to 5%) which agrees with CMAM as well. The CMAM EPP energy deposition was nonlocal extending below 80 km in a rapidly evanescent tail unlike in EMAC where an upper boundary condition was used. So it appears consistent with the vertical transport in EMAC being hyperactive at least in the mesosphere.

However, it is not clear that electron EPP is zero below 80 km during winter. In fact, there is likely to be a substantial energy deposition between 70 and 80 km associated with relativistic electrons. Differences in transport above 80 km between the two high lid models likely account for the large NOy differences compared to MIPAS. In some sense the boundary condition approach in low-lid EMAC may be better since the EPP NOy (and HOx) production is not as affected by model transport pathologies above 80 km. Both KASIMA and EMAC have better NOy distribution patterns in the upper meso-sphere compared ot 3dCTM (e.g. 2003-4, 2009-10 in both hemisphere) but EMAC appears to be best.

The differences in terms of ozone loss shown in Figures 8 and 9 do not necessarily reflect problems with EMAC. There is sufficient reason to question the realism of ACPD

---

## Referee Comment (RC2) · Anonymous Referee #2 · 14 Jul 2017

Overall this is an interesting, well-conceived and well written paper that should be of solid interest for ACP readers. I have one general concern. Assuming it can be addressed, along with several more minor concerns, the paper should be suitable for publication.

The main concern is that one can't help but note that the authors put much more effort into validating their NOy calculations by comparing continuously with MIPAS. Section 3.3 is quite good in this regard. They do not do this with ozone; rather, they reference other work. However, this is unsatisfactory and the net effect is that their modeled ozone changes are less robust than their modeled NOy changes. Particularly for some

of the strongest years, like 2003-2004, it would be extremely helpful and much more convincing just to show their calculated ozone for with and without EPP/SPE compared with some observations. Even simple comparisons- like for example, the ozone reduction discussed by Natarajan et al [2004, GRL] (and which remains, in my opinion, the most compelling case of stratospheric ozone reductions in response to these events, and which unfortunately, is not cited) would be better than nothing. Although it would be hoped that they could do more. Many previous works (Siskind, Funke, Jackman) gave the various contributions to NOy in both absolute numbers but also percent. Given the differing transport amongst the 3 models, it would be useful to get percent contributions. This would be especially helpful for the ozone and heating rate calculations. I had a very hard time deciding how significant these effects were. Continuing on the above thread, for Figure 11- the post SSW changes look interesting. Again, along the lines of my comments above, it would be much more compelling if they could do some comparison with observations. For example, show an average of the 3 post-SSW year temperatures compared with non-SSW years and then with their model. Given how comprehensive their NOy model-data comparisons were, the lack of such comparisons for ozone and heating/temperature changes are more apparent. Ultimately, with the uncertainty both in relative contribution and as well as validation, I find the last several lines (12-15 on page 32) to be too speculative in my opinion and should be removed. I only see a significant blob of color on the EMAC plot and only for one spring. Almost nothing in 2006 and 2009.

Minor Intro: I believe SPEs have been known to be sufficiently energetic to directly ionize the lower stratosphere. The text keeps saying "upper". English grammar: line 28 page 5 "prevents .. from propagating" Figure 2- they should zoom in on the vertical axis- there is no reason to show 5 orders of magnitude when 3 will cover the range. Line 9, page 6; line 1 page 10 and many places elsewhere. Could the authors please use hPa rather than Pa? Unless ACP has a preference, I believe more people in aeronomy intuitively think in terms of hPa. Related to the above- line 29, page 12. Doesn't > 100 pa refer to the stratosphere? The text says mesosphere. Figure 8-

why are there apparent vertical discontinuities in the ozone change? For example, top panel, beginning of 2005 where the colors go from dark green to blue instantly over a wide range of values.

Referencing: I believe Randall et al 2007 first coined the phrase "EPP-IE". They should be cited on line 8 of page 3. They already are cited elsewhere.

Siskind references get the first initials wrong. In one place its DR in another its DW. Should be DE

Same problem with Fleimng- its EK in the reference with Jackman. EL in other places.

Matthes paper is 2017 as it has appeared now.

---

## Short Comment (SC1) · 28 Jul 2017

Comment considering the conclusions of the paper "NOy production, ozone loss and changes in net radiative heating due to energetic particle precipitation in 2002–2010" by Sinnhuber et al.

The paper states that "Analysis of several decades of re-analysis data show a warming of the mid-to late winter upper stratosphere related to high geomagnetic activity (Lu et al., 2008; Seppälä et al., 2013). These have been interpreted as a result of coupling between the vortex strength and wave propagation and reflection, an assumption strengthened by the apparent relation to the phase of the stratospheric quasi-biennal

oscillation and the solar cycle (Lu et al., 2008; Seppälä et al., 2013). However, our results suggest that the direct radiative impact plays a role as well."

As we write (about the upper stratospheric warming signal) in Seppälä et al., 2013, the paper referenced here: "Based on earlier work by others, Baumgaertner et al. [2011] suggested that the warming signal would be a result in decrease in ozone radiative cooling as a response to ozone depletion, and the cooling signal might arise from dynamical heating due to slowing down of the meridional Brewer-Dobson circulation. Such a reduction would be associated with less upward EP flux and more waves reflecting toward the equator [see Lu et al., 2008b, and references therein]. As discussed above, this is now confirmed by our EP flux results." (Discussion, last paragraph)

"These results confirm the previous hypothesis of Lu et al. [2008b] regarding the role of dynamics in coupling geomagnetic activity levels and stratospheric changes and supports the suggestion of Baumgaertner et al. [2011] about the dynamical coupling mechanism connecting EPP-NOx induced ozone loss, polar stratospheric temperatures, and the modulation of the Northern Annular Mode." (Conclusions, last paragraph)

This clearly shows that the role of the direct radiative impact in the upper stratosphere suggested here is in no disagreement by the works cited. Results of Lu et al. (2008) and Seppälä et al. (2013) both support the suggestion of Baumgaertner et al. (2011) (which is unfortunately not cited in the current paper in this context) that the upper stratosphere warming signal is related to the radiative changes arising from changes in ozone, exactly as proposed again here.

In summary, the results of the modelling work done here regarding the role of direct radiative impact do, in fact, strongly support the results of Lu et al. (2008), Baumgaertner et al. (2011) and Seppälä et al (2013). This agreement between the independent studies is important, and I would be grateful if the authors would correct their statement concerning this in the current paper.

---

## Author Comment (AC1) · 22 Sep 2017

We would like to thank the reviewer for the careful review of the paper, which certainly has made the paper more interesting.

As suggested by reviewer 2, pressure is now given in hPa, not in Pa. Figures, tables and text have been redone accordingly. A bug in the zonal averaging of the 3dCTM data has been corrected. This has no impact on the conclusions, but leads to slightly higher NO$_y$ (figures 2, 4, and 6, particularly after the SSW in early 2009) and ozone losses (former figure 9) in the Northern hemisphere. Results shown in former figures 7, table 2, and former figures 10 and 11 are not affected as those were averaged correctly.

Following the suggestion of Reviewer 2, new figures have been included showing a comparison to MIPAS ozone observations, and a comparison to (modeled) net radiative heating rates. New references have been included: Semeniuk et al. (2011); Natarajan et al. (2004); Randall et al. (2005); B-M. Sinnhuber et al. (2002); Damiani et al. (2016).

*On page 26 (around line 20) the authors speculate that application of the boundary condition in EMAC at 80 km may be resulting in excessive ozone loss. Semeniuk et al. (2011) using CMAM simulated large ozone losses around 50 % at 80 km in the winter polar caps (70-90) South and North due to EPP HOx. Peak loss values were over 70 % which agrees with the EMAC predicted losses averaged over the same period. CMAM did not use a boundary condition to drive EPP NOx but used the Jackman scheme for EPP NOx production and the Solomon scheme for EPP HOx production. So author's speculation about excessive O3 destruction based on excessive activity of the NO + O3 reaction does not apply to the CMAM and hence appears to be dubious. I suggest the authors either change the discussion of this aspect or conduct a more detailed chemistry analysis.*

Thanks for pointing this out. A comparison of the modeled ozone loss with the CMAM results in the stratosphere and mesosphere has been added to the paper. A reference to (Semeniuk et al., 2011) has been added to the introduction, to the discussion of the stratospheric and mesospheric ozone loss in former Sec. 4.1, and also to the discussion of temperature changes at the stratopause in the Conclusions.

The discussion of the strong mesospheric ozone loss in the EMAC model has been adapted to include the reaction NO + $HO_2 \longrightarrow NO_2$ + OH, which re-partitions HOx, leading to an increase in HOx and enhanced mesospheric ozone loss as shown in (Verronen and Lehmann, 2015). However, we would like to point out the following: the increase of mesospheric HOx due to particle impact ionization is considered in EMAC only for solar proton events, not for medium-energy electrons. The strong ozone loss in EMAC therefore is not comparable to the CMAM results (or, to put it another way, mesospheric ozone loss in EMAC likely agrees well with CMAM for the wrong reasons).

The strong mesospheric ozone loss in EMAC is an artefact of the method of prescribing NOx in the form of NO.

*At the same time the ozone loss in the stratosphere aside from the peak years (2004-2005, 2006-2007) in the northern hemisphere is below 10 % (closer to 5 %) which agrees with CMAM as well. The CMAM EPP energy deposition was nonlocal extending below 80 km in a rapidly evanescent tail unlike in EMAC where an upper boundary condition was used. So it appears consistent with the vertical transport in EMAC being hyperactive at least in the mesosphere.*

A comparison with CMAM results has been added to the discussion of the stratospheric and mesospheric ozone loss.

*However, it is not clear that electron EPP is zero below 80 km during winter. In fact, there is likely to be a substantial energy deposition between 70 and 80 km associated with relativistic electrons.*

Yes, and in 3dCTM and KASIMA, particle precipitation is precribed throughout the stratosphere and mesosphere. As the upper boundary condition used in EMAC is based on observations of mesospheric NOy, it implicitly includes the direct impact of particle precipitation into the mesosphere; the observations cannot separate between direct and indirect impacts. However, the direct impact of particle precipitation on HOx is included in EMAC only during solar proton events.

*Differences in transport above 80 km between the two high lid models likely account for the large NOy differences compared to MIPAS. In some sense the boundary condition approach in low-lid EMAC may be better since the EPP NOy (and HOx) production is not as affected by model transport pathologies above 80 km. Both KASIMA and EMAC have better NOy distribution patterns in the upper mesosphere compared to 3dCTM (e.g. 2003-4, 2009-10 in both hemisphere) but EMAC appears to be best.*

Yes. E.g., further experiments with KASIMA show that the upper boundary condition at

120 km affects the NOy budget at 80 km. Fixing the NOy value to a parameterization of observation avoids such problems. In this sense and not unexpected, the approach used in the EMAC simulation is perhaps more appropriate to assess the NOy impact in the MA.

*The differences in terms of ozone loss shown in Figures 8 and 9 do not necessarily reflect problems with EMAC. There is sufficient reason to question the realism of chemical conditions in the upper mesosphere in 3dCTM and KASIMA.*

The method of implementing the impact of auroral and medium-energy electrons in EMAC by the upper boundary condition is unrealistic for two reasons: a) HOx production due to medium-energy electrons in the upper mesosphere (70–80 km) is not considered in the model, and b) NOy is precribed in the form of NO, which probably leads to an unrealistic partitioning of both NOx and HOx; particularly the partitioning of HOx has a big impact on ozone as discussed by Verronen and Lehmann (2015). It is possible that the change in HOx partitioning partly balances the missing EPP HOx; however, in this case, ozone in EMAC may be correct, but for the wrong reasons.

**References**

Damiani, A., Funke, B., López-Puertas, M., Santee, M.L., Cordero, R.R., and Watanabe, S., Energetic particle precipitation: a major driver of the ozone budget in the Antarctic upper stratosphere, Geophys. Res. Lett., 43m 3554–3562, doi: 10.1002/2016GL068279, 2016.

Natarajan, M., Remsberg, E.E., and Deaver, L.E., Anomalously high levels of NOx in the polar upper stratosphere during April, 2004: Photochemical consistency of HALOE observations, Geophys. Res. Lett., 31, doi: 10.1029/2004GL020566, 2004.

Randall, C.E., Harvey, V.L. Manney, G.L., Orsolini, Y., Codrescu, M., Sioris, C., Brohede, S., Haley, C.S., Gordley, L.L., Zawodny, J.M., and Russell, J.M. III, Stratospheric effects of energetic particle precipitation in 2003–2004, Geophys. Res. Lett., 32, doi: 10.1029/2004GL022003, 2005.

Randall, C.E., Harvey, V.L., Singleton, C.S., Bailey, S.M., Bernath, P.F., Codrescu, M., Nakajima, H. and Russell, J.M. III, Energetic particle precipitation effects on the Southern Hemisphere stratosphere in 1992–2005, J. Geophys. Res., 112, doi:10.1029/2006JD007696, 2007.

Reddmann, T., Ruhnke, R., Versick, S., and Kouker, W.: Modeling disturbed stratospheric chemistry during solar-induced NOx enhancements observed with MIPAS/ENVISAT, J. Geophys. Res., 115, doi:10.1029/2009JD012569, 2010.

Semeniuk, K., Fomichev, V.I., McConnell, J.C., Fu, C., Melo, S.M.L., and Usoskin, I.G., Middle atmosphere response to the solar cycle in i rradiance and ionizing particle impact, Atmos. Chem. Phys., 11, 5045–5077, doi:10.5194/acp-11-5045-2011, 2011.

Sinnhuber, B.-M., Weber, M., Amankwah, A., and Burrows, J.P., Total ozone during the unusual Antarctic winter of 2002, Geophys. Res. Lett., 30, 1580, doi: 10.1029/2002GL016798, 2002.

Verronen, P.T., and Lehmann, R., Enhancement of odd nitrogen modifies mesospheric ozone chemistry during polar winter, Geophys. Res. Lett., 42, 10445–10452, doi:10.1002/2015GL066703, 2015.

---

## Author Comment (AC3) · 22 Sep 2017

*Overall this is an interesting, well-conceived and well written paper that should be of solid interest for ACP readers. I have one general concern. Assuming it can be addressed, along with several more minor concerns, the paper should be suitable for publication.*

We would like to thank the reviewer for this opinion, and for the careful review which certainly has made the paper more interesting.

As suggested, pressure is now given in hPa, not in Pa. Figures, tables and text have

been redone accordingly. A bug in the zonal averaging of the 3dCTM data has been corrected. This has no impact on the conclusions, but leads to slightly higher $NO_y$ (figures 2, 4, and 6, particularly after the SSW in early 2009) and ozone losses (former figure 9) in the Northern hemisphere. Results shown in former figures 7, table 2, and former figures 10 and 11 are not affected as those were averaged correctly. Following the suggestion of the reviewer, new figures have been included showing a comparison to MIPAS ozone observations, and a comparison to (modeled) net radiative heating rates. New references have been included: Natarajan et al. (2004); Randall et al. (2005); B-M. Sinnhuber et al. (2002); Damiani et al. (2016); Semeniuk et al. (2011).

*The main concern is that one can't help but note that the authors put much more effort into validating their NOy calculations by comparing continuously with MIPAS. Section 3.3 is quite good in this regard. They do not do this with ozone; rather, they reference other work. However, this is unsatisfactory and the net effect is that their modeled ozone changes are less robust than their modeled NOy changes.*

Thanks for pointing this out. We have added a comparison of model results with MIPAS ozone. A new subsection Sec. 4.1 *Comparison of modeled and observed ozone fields* is added at the beginning of Section 4. Figures comparing MIPAS ozone to model results from the EPP runs have been included, see figures Fig. 1 and Fig. 2.

*Particularly for some of the strongest years, like 2003-2004, it would be extremely helpful and much more convincing just to show their calculated ozone for with and without EPP/SPE compared with some observations.*

Different approaches to this have been tested. It turned out to be difficult to extract comparable results over a larger period of time and/or vertical range because of the strong dynamical variability of ozone, which is notably larger than the ozone variability due to EPP in most years. Two new figures (figures Fig. 3 and Fig. 4) were included comparing relative ozone anomalies, that is the (relative) difference between daily values and a multi-annual mean. The period 2006-2009 was chosen for the multi-annual

mean because it is covered well by MIPAS observations, and is characterized by relatively low geomagnetic activity. Model results for both the EPP and the Base model scenarios are shown for the period 2002.5 to 2004.5, when significant differences between the model runs with and without EPP are observed in both hemispheres in the mesosphere and upper stratosphere, particularly after the SSW in the Northern hemisphere (EMAC). This has been added in the new subsection Sec. 4.2 *Comparison of modeled and observed ozone anomalies*.

*Even simple comparisons- like for example, the ozone reduction discussed by Natarajan et al [2004, GRL] (and which remains, in my opinion, the most compelling case of stratospheric ozone reductions in response to these events, and which unfortunately, is not cited) would be better than nothing. Although it would be hoped that they could do more.*

Thanks for pointing this out. A reference to Natarajan et al. (2004) and Randall et al. (2005) has been added to the introduction. The results of Natarajan et al. (2004) and Randall et al. (2005) as well as model results from (Semeniuk et al., 2011; Baumgaertner et al., 2011) are also discussed in comparison to the modeled ozone differences in Sec. 3.3 (previous Sec. 3.1).

*Many previous works (Siskind, Funke, Jackman) gave the various contributions to NOy in both absolute numbers but also percent. Given the differing transport amongst the 3 models, it would be useful to get percent contributions. This would be especially helpful for the ozone and heating rate calculations. I had a very hard time deciding how significant these effects were.*

We have added absolute values to the ozone differences and added some discussion about how these absolute values compare to observations (Natarajan et al., 2004; Randall et al., 2005) and other model studies (Reddmann et al., 2010). We did not add relative differences to NOy. These were tested for Figures 5 and 6, and it was found that the general conclusions do not change if relative differences are given. Absolute

differences cover two orders of magnitudes (from 2 ppb to more than 100 ppb), and relative differences of more than 100 % therefore occur related to differences in the speed of the downwelling of the EPP $NO_y$ signal. However, large percentage changes also occur in regions of very low $NO_y$, e.g., at the edge of the Antarctic ozone hole area where $NO_y$ is depleted. For the heating rates, the net radiative heating rates of the Base runs of EMAC were added for comparison to the bottom panels of former figures Fig. 10 and 11 (now Figs. 14 and 15).

*Continuing on the above thread, for Figure 11- the post SSW changes look interesting.*

Comparisons with the net radiative heating rates of the EMAC model show that the largest changes to the heating rates, during spring in the upper stratosphere after large wintertime solar proton events or sudden stratospheric warmings, occur when the atmosphere approaches radiative equililbrium, i.e., net radiative heating rates are low (less than 0.5 K/day to 1–2.5 K/day), see lower panels of Figs. 5 and 6. Relative changes can approach 100 %, and potentially have a bigger impact than the changes during midwinter, when the net radiative heating rates are higher. During midwinter, relative changes are in the range of 2–10 % of the net value only. A more detailed discussion of this has been added to the text.

*Again, along the lines of my comments above, it would be much more compelling if they could do some comparison with observations. For example, show an average of the 3 post-SSW year temperatures compared with non-SSW years and then with their model. Given how comprehensive their NOy model-data comparisons were, the lack of such comparisons for ozone and heating/temperature changes are more apparent.*

This is a good suggestion; however, it is not possible to carry it out with the model scenarios as set up here. All model runs shown in this paper are done with a specified dynamics setup, meaning that in the stratosphere and troposphere, temperatures and wind-fields are relaxed to observations. This is done in the same way for the EPP and Base model runs, so stratospheric temperatures are almost identical. This model

setup was chosen to ensure that the interannual variation of ozone and NOy agree with observations of specific years as well as possible. For a study of the impact of the heating rates on stratospheric temperatures, follow-up studies with free-running models are needed. These would be of great interest obviously, but are out of the scope of this paper.

*Ultimately, with the uncertainty both in relative contribution and as well as validation, I find the last several lines (12-15 on page 32) to be too speculative in my opinion and should be removed. I only see a significant blob of color on the EMAC plot and only for one spring. Almost nothing in 2006 and 2009.*

We agree that this should have been formulated (and now is formulated) more carefully:

*The indirect effect could contribute to the reformation of a strong and long-lasting vortex in late winter and spring after sudden stratospheric warmings in years with high geomagnetic activity, as e.g. in winter 2003–2004.*

Though this is observed only in the EMAC model, mentioning it seems justified to us because the impact of sudden stratospheric warmings on both NOy and ozone are better represented by EMAC than by the other two models.

*Intro: I believe SPEs have been known to be sufficiently energetic to directly ionize the lower stratosphere. The text keeps saying "upper".*

A note in the introduction was added that protons can penetrate down to the lower stratosphere in events with particularly hard energy spectra.

*English grammar: line 28 page 5 "prevents .. from propagating"*

Changed.

*Figure 2- they should zoom in on the vertical axis- there is no reason to show 5 orders of magnitude when 3 will cover the range.*

Done.

*Line 9, page 6; line 1 page 10 and many places elsewhere. Could the authors please use hPa rather than Pa? Unless ACP has a preference, I believe more people in aeronomy intuitively think in terms of hPa.*

Done.

*Related to the above- line 29, page 12. Doesn't > 100 pa refer to the stratosphere? The text says mesosphere.*

The sign was reversed, it now says ≤ 1 hPa.

*Figure 8- why are there apparent vertical discontinuities in the ozone change? For example, top panel, beginning of 2005 where the colors go from dark green to blue instantly over a wide range of values.*

This is the direct impact of solar proton events, e.g., in October 2003, January 2005, and December 2006. In the model results, SPEs deplete ozone in the whole vertical range from about 10 hPa (30 km) up to the mesopause; however, it should be pointed out that these changes related to strong solar proton events in the mid-to-upper stratosphere are a few percent only, probably too low to be observable. A bullet point discussing the solar proton event impact in more detail has been added to the discussion.

*Referencing: I believe Randall et al 2007 first coined the phrase "EPP-IE". They should be cited on line 8 of page 3. They already are cited elsewhere.*

Thanks for pointing this out, the reference to Randall et al. (2007) has been added at this point.

*Siskind references get the first initials wrong. In one place its DR in another its DW. Should be DE*

Changed.

*Same problem with Fleimng- its EK in the reference with Jackman. EL in other places.*

Changed to the correct (EL).

*Matthes paper is 2017 as it has appeared now.*

Changed.

**References**

Baumgaertner, A.J.G., Seppälä, A., Jöckel, P., and Clilverd, M.A., Geomagnetic activity related $NO_x$ enhancements and polar surface temperature variability in a chemistry climate model: modulation of the NAM index, Atmos. Chem. Phys., 11, 4521–4531, doi:10.5194/acp-11-4521-2011, 2011.

Damiani, A., Funke, B., López-Puertas, M., Santee, M.L., Cordero, R.R., and Watanabe, S., Energetic particle precipitation: a major driver of the ozone budget in the Antarctic upper stratosphere, Geophys. Res. Lett., 43m 3554–3562, doi: 10.1002/2016GL068279, 2016.

Natarajan, M., Remsberg, E.E., and Deaver, L.E., Anomalously high levels of NOx in the polar upper stratosphere during April, 2004: Photochemical consistency of HALOE observations, Geophys. Res. Lett., 31, doi: 10.1029/2004GL020566, 2004.

Randall, C.E., Harvey, V.L. Manney, G.L., Orsolini, Y., Codrescu, M., Sioris, C., Brohede, S., Haley, C.S., Gordley, L.L., Zawodny, J.M., and Russell, J.M. III, Stratospheric effects of energetic particle precipitation in 2003–2004, Geophys. Res. Lett., 32, doi: 10.1029/2004GL022003, 2005.

Randall, C.E., Harvey, V.L., Singleton, C.S., Bailey, S.M., Bernath, P.F., Codrescu, M., Nakajima, H. and Russell, J.M. III, Energetic particle precipitation effects on the Southern Hemisphere stratosphere in 1992–2005, J. Geophys. Res., 112, doi:10.1029/2006JD007696, 2007.

Reddmann, T., Ruhnke, R., Versick, S., and Kouker, W.: Modeling disturbed stratospheric chemistry during solar-induced NOx enhancements observed with MIPAS/ENVISAT, J. Geophys. Res., 115, doi:10.1029/2009JD012569, 2010.

Semeniuk, K., Fomichev, V.I., McConnell, J.C., Fu, C., Melo, S.M.L., and Usoskin, I.G., Middle atmosphere response to the solar cycle in irradiance and ionizing particle impact, Atmos. Chem. Phys., 11, 5045–5077, doi:10.5194/acp-11-5045-2011, 2011.

Sinnhuber, B.-M., Weber, M., Amankwah, A., and Burrows, J.P., Total ozone during the unusual Antarctic winter of 2002, Geophys. Res. Lett., 30, 1580, doi: 10.1029/2002GL016798, 2002.
* * *
MIPAS O$_3$ [ppm], SH

3dCTM v1.6 phioniz - MIPAS O$_3$ [ppm], SH

KASIMA v1.6 - MIPAS O$_3$ [ppm], SH

EMAC UBC - MIPAS O$_3$ [ppm], SH

**Fig. 1.**

MIPAS O$_3$ [ppm], NH

3dCTM v1.6 phioniz - MIPAS O$_3$ [ppm], NH

KASIMA v1.6 - MIPAS O$_3$ [ppm], NH

EMAC UBC - MIPAS O$_3$ [ppm], NH

**Fig. 2.**

[Figure]

[Figure]

[Figure]

**Fig. 3.**

O3 MIPAS - climatology, %, NH

O3 3dCTM v1.6 phioniz - climatology, %, NH

O3 3dCTM Base - climatology, %, NH

O3 KASIMA v1.6 - climatology, %, NH

O3 KASIMA Base - climatology, %, NH

O3 EMAC UBC - climatology, %, SH

O3 EMAC Base - climatology, %, SH

**Fig. 4.**

![3dCTM, d(dT/dt) (IR+UV) [K/day], 70S-90S; KASIMA, d(dT/dt) (IR+UV) [K/day], 70S-90S; EMAC, d(dT/dt) (IR+UV) [K/day], 70S-90S; EMAC Base, net radiative heating rate [K/day], 70S-90S]()

**Fig. 5.**

**3dCTM, d(dT/dt) (IR+UV) [K/day], 70N-90N**

**KASIMA, d(dT/dt) (IR+UV) [K/day], 70N-90N**

**EMAC, d(dT/dt) (IR+UV) [K/day], 70N-90N**

**EMAC Base, net radiative heating rate [K/day], 70N-90N**

**Fig. 6.**

---

## Author Response (AR1)

**Reply to reviewer 1**

We would like to thank the reviewer for the careful review of the paper, which certainly has made the paper more interesting. As suggested by reviewer 2, pressure is now given in hPa, not in Pa. Figures, tables and text have been redone accordingly. A bug in the zonal averaging of the 3dCTM data has been corrected. This has no impact on the conclusions, but leads to slightly higher $NO_y$ (figures 2, 4, and 6, particularly after the SSW in early 2009) and ozone losses (former figure 9) in the Northern hemisphere. Results shown in former figures 7, table 2, and former figures 10 and 11 are not affected as those were averaged correctly. Following the suggestion of Reviewer 2, new figures have been included showing a comparison to MIPAS ozone observations, and a comparison to (modeled) net radiative heating rates. New references have been included: Semeniuk et al. (2011); Natarajan et al. (2004); Randall et al. (2005); B-M. Sinnhuber et al. (2002); Damiani et al. (2016).

*On page 26 (around line 20) the authors speculate that application of the boundary condition in EMAC at 80 km may be resulting in excessive ozone loss. Semeniuk et al. (2011) using CMAM simulated large ozone losses around 50 % at 80 km in the winter polar caps (70-90) South and North due to EPP HOx. Peak loss values were over 70 % which agrees with the EMAC predicted losses averaged over the same period. CMAM did not use a boundary condition to drive EPP NOx but used the Jackman scheme for EPP NOx production and the Solomon scheme for EPP HOx production. So author's speculation about excessive O3 destruction based on excessive activity of the NO + O3 reaction does not apply to the CMAM and hence appears to be dubious. I suggest the authors either change the discussion of this aspect or conduct a more detailed chemistry analysis.*

Thanks for pointing this out. A comparison of the modeled ozone loss with the CMAM results in the stratosphere and mesosphere has been added to the paper. A reference to (Semeniuk et al., 2011) has been added to the introduction, to the discussion of the stratospheric and mesospheric ozone loss in former Sec. 4.1, and also to the discussion of temperature changes at the stratopause in the Conclusions.

The discussion of the strong mesospheric ozone loss in the EMAC model has been adapted to include the reaction $NO + HO_2 \longrightarrow NO_2 + OH$, which re-partitions HOx, leading to an increase in HOx and enhanced mesospheric ozone loss as shown in (Verronen and Lehmann, 2015). However, we would like to point out the following: the increase of mesospheric HOx due to particle impact ionization is considered in EMAC only for solar proton events, not for medium-energy electrons. The strong ozone loss in EMAC therefore is not comparable to the CMAM results (or, to put it another way, mesospheric ozone loss in EMAC likely agrees well with CMAM for the wrong reasons). The strong mesospheric ozone loss in EMAC is an artefact of the method of prescribing NOx in the form of NO.

*At the same time the ozone loss in the stratosphere aside from the peak years (2004-2005, 2006-2007) in the northern hemisphere is below 10 % (closer to 5 %) which agrees with CMAM as well. The CMAM EPP energy deposition was nonlocal extending below 80 km in a rapidly evanescent tail unlike in EMAC where an upper boundary condition was used. So it appears consistent with the vertical transport in EMAC being hyperactive at least in the mesosphere.*

A comparison with CMAM results has been added to the discussion of the stratospheric and mesospheric ozone loss.

*However, it is not clear that electron EPP is zero below 80 km during winter. In fact, there is likely to be a substantial energy deposition between 70 and 80 km associated with relativistic electrons.*

Yes, and in 3dCTM and KASIMA, particle precipitation is precribed throughout the stratosphere and mesosphere. As the upper boundary condition used in EMAC is based on observations of mesospheric $NO_y$, it implicitly includes the direct impact of particle precipitation into the mesosphere; the observations cannot separate between direct and indirect impacts. However, the direct impact of particle precipitation on HOx is included in EMAC only during solar proton events.

*Differences in transport above 80 km between the two high lid models likely account for the large NOy differences compared to MIPAS. In some sense the boundary condition approach in low-lid EMAC may be better since the EPP NOy (and HOx) production is not as affected by model transport pathologies above 80 km. Both KASIMA and EMAC have better NOy distribution patterns in the upper mesosphere compared to 3dCTM (e.g. 2003-4, 2009-10 in both hemisphere) but EMAC appears to be best.*

Yes. E.g., further experiments with KASIMA show that the upper boundary condition at 120 km affects the NOy budget at 80 km. Fixing the NOy value to a parameterization of observation avoids such problems. In this sense and not unexpected, the approach used in the EMAC simulation is perhaps more appropriate to assess the NOy impact in the MA.

*The differences in terms of ozone loss shown in Figures 8 and 9 do not necessarily reflect problems with EMAC. There is sufficient reason to question the realism of chemical conditions in the upper mesosphere in 3dCTM and KASIMA.*

The method of implementing the impact of auroral and medium-energy electrons in EMAC by the upper boundary condition is unrealistic for two reasons: a) HOx production due to medium-energy electrons in the upper mesosphere (70–80 km) is not considered in the model, and b) NOy is prescribed in the form of NO, which probably leads to an unrealistic partitioning of both NOx and HOx; particularly the partitioning of HOx has a big impact on ozone as discussed by Verronen and Lehmann (2015).
It is possible that the change in HOx partitioning partly balances the missing EPP HOx; however, in this case, ozone in EMAC may be correct, but for the wrong reasons.

**Reply to reviewer 2**

*Overall this is an interesting, well-conceived and well written paper that should be of solid interest for ACP readers. I have one general concern. Assuming it can be addressed, along with several more minor concerns, the paper should be suitable for publication.*

We would like to thank the reviewer for this opinion, and for the careful review which certainly has made the paper more interesting.

As suggested, pressure is now given in hPa, not in Pa. Figures, tables and text have been redone accordingly. A bug in the zonal averaging of the 3dCTM data has been corrected. This has no impact on the conclusions, but leads to slightly higher $NO_y$ (figures 2, 4, and 6, particularly after the SSW in early 2009) and ozone losses (former figure 9) in the Northern hemisphere. Results shown in former figures 7, table 2, and former figures 10 and 11 are not affected as those were averaged correctly. Following the suggestion of the reviewer, new figures have been included showing a comparison to MIPAS ozone observations, and a comparison to (modeled) net radiative heating rates. New references have been included: Natarajan et al. (2004); Randall et al. (2005); B-M. Sinnhuber et al. (2002); Damiani et al. (2016); Semeniuk et al. (2011).

*The main concern is that one can't help but note that the authors put much more effort into validating their NOy calculations by comparing continuously with MIPAS. Section 3.3 is quite good in this regard. They do not do this with ozone; rather, they reference other work. However, this is unsatisfactory and the net effect is that their modeled ozone changes are less robust than their modeled NOy changes.*

Thanks for pointing this out. We have added a comparison of model results with MIPAS ozone. A new subsection Sec. 4.1 *Comparison of modeled and observed ozone fields* is added at the beginning of Section 4. Figures comparing MIPAS ozone to model results from the EPP runs have been included, see figures Fig. 1 and Fig. 2.

*Particularly for some of the strongest years, like 2003-2004, it would be extremely helpful and much more convincing just to show their calculated ozone for with and without EPP/SPE compared with some observations.*

Different approaches to this have been tested. It turned out to be difficult to extract comparable results over a larger period of time and/or vertical range because of the strong dynamical variability of ozone, which is notably larger than the ozone variability due to EPP in most years. Two new figures (figures Fig. 3 and Fig. 4) were included comparing relative ozone anomalies, that is the (relative) difference between daily values and a multi-annual mean. The period 2006-2009 was chosen for the multi-annual mean because it is covered well by MIPAS observations, and is characterized by relatively low geomagnetic activity. Model results for both the EPP and the Base model scenarios are shown for the period 2002.5 to 2004.5, when significant differences between the model runs with and without EPP are observed in both hemispheres in the mesosphere and upper stratosphere, particularly after the SSW in the Northern hemisphere (EMAC). This has been added in the new subsection Sec. 4.2 *Comparison of modeled and observed ozone anomalies*.

*Even simple comparisons- like for example, the ozone reduction discussed by Natarajan et al [2004, GRL] (and which remains, in my opinion, the most compelling case of stratospheric ozone reductions in response to these events, and which unfortunately, is not cited) would be better than nothing. Although it would be hoped that they could do more.*

Thanks for pointing this out. A reference to Natarajan et al. (2004) and Randall et al. (2005) has been added to the introduction. The results of Natarajan et al. (2004) and Randall et al. (2005) as well as model results from (Semeniuk et al., 2011; Baumgaertner et al., 2011) are also discussed in comparison to the modeled ozone differences in Sec. 3.3 (previous Sec. 3.1).

*Many previous works (Siskind, Funke, Jackman) gave the various contributions to NOy in both absolute numbers but also percent. Given the differing transport amongst the 3 models, it would be useful to get percent contributions. This would be especially helpful for the ozone and heating rate calculations. I had a very hard time deciding how significant these effects were.*

We have added absolute values to the ozone differences and added some discussion about how these absolute values compare to observations (Natarajan et al., 2004; Randall et al., 2005) and other model studies (Reddmann et al., 2010). We did not add relative differences to NOy. These were tested for Figures 5 and 6, and it was found that the general conclusions do not change if relative differences are given. Absolute differences cover two orders of magnitudes (from 2 ppb to more than 100 ppb), and relative differences of more than 100 % therefore occur related to differences in the speed of the downwelling of the EPP $NO_y$ signal. However, large percentage changes also occur in regions of very low $NO_y$, e.g., at the edge of the Antarctic ozone hole area where $NO_y$ is depleted. For the heating rates, the net radiative heating rates of the Base runs of EMAC were added for comparison to the bottom panels of former figures Fig. 10 and 11 (now Figs. 14 and 15).

*Continuing on the above thread, for Figure 11- the post SSW changes look interesting.*

Comparisons with the net radiative heating rates of the EMAC model show that the largest changes to the heating rates, during spring in the upper stratosphere after large wintertime solar proton events or sudden stratospheric warmings, occur when the atmosphere approaches radiative equilibrium, i.e., net radiative heating rates are low (less than 0.5 K/day to 1– 2.5 K/day), see lower panels of Figs. 5 and 6. Relative changes can approach 100 %, and potentially have a bigger impact than the changes during midwinter, when the net radiative heating rates are higher. During midwinter, relative changes are in the range of 2–10 % of the net value only. A more detailed discussion of this has been added to the text.

*Again, along the lines of my comments above, it would be much more compelling if they could do some comparison with observations. For example, show an average of the 3 post-SSW year temperatures compared with non-SSW years and then with their model. Given how comprehensive their NOy model-data comparisons were, the lack of such comparisons for ozone and heating/temperature changes are more apparent.*

This is a good suggestion; however, it is not possible to carry it out with the model scenarios as set up here. All model runs shown in this paper are done with a specified dynamics setup, meaning that in the stratosphere and troposphere, temperatures and wind-fields are relaxed to observations. This is done in the same way for the EPP and Base model runs, so stratospheric temperatures are almost identical. This model setup was chosen to ensure that the interannual variation of ozone and NOy agree with observations of specific years as well as possible. For a study of the impact of the heating rates on stratospheric temperatures, follow-up studies with free-running models are needed. These would be of great interest obviously, but are out of the scope of this paper.

*Ultimately, with the uncertainty both in relative contribution and as well as validation, I find the last several lines (12-15 on page 32) to be too speculative in my opinion and should be removed. I only see a significant blob of color on the EMAC plot and only for one spring. Almost nothing in 2006 and 2009.*

We agree that this should have been formulated (and now is formulated) more carefully: *The indirect effect could contribute to the reformation of a strong and long-lasting vortex in late winter and spring after sudden stratospheric warmings in years with high geomagnetic activity, as e.g. in winter 2003–2004.* Though this is observed only in the EMAC model, mentioning it seems justified to us because the impact of sudden stratospheric warmings on both NOy and ozone are better represented by EMAC than by the other two models.

*Intro: I believe SPEs have been known to be sufficiently energetic to directly ionize the lower stratosphere. The text keeps saying "upper".*

A note in the introduction was added that protons can penetrate down to the lower stratosphere in events with particularly hard energy spectra.

*English grammar: line 28 page 5 "prevents .. from propagating"*

Changed.

*Figure 2- they should zoom in on the vertical axis- there is no reason to show 5 orders of magnitude when 3 will cover the range.*

Done.

*Line 9, page 6; line 1 page 10 and many places elsewhere. Could the authors please use hPa rather than Pa? Unless ACP has a preference, I believe more people in aeronomy intuitively think in terms of hPa.*

Done.

*Related to the above- line 29, page 12. Doesn't > 100 pa refer to the stratosphere? The text says mesosphere.*

The sign was reversed, it now says $\leq 1$ hPa.

*Figure 8- why are there apparent vertical discontinuities in the ozone change? For example, top panel, beginning of 2005 where the colors go from dark green to blue instantly over a wide range of values.*

This is the direct impact of solar proton events, e.g., in October 2003, January 2005, and December 2006. In the model results, SPEs deplete ozone in the whole vertical range from about 10 hPa (30 km) up to the mesopause; however, it should be pointed out that these changes related to strong solar proton events in the mid-to upper stratosphere are a few percent only, probably too low to be observable. A bullet point discussing the solar proton event impact in more detail has been added to the discussion.

*Referencing: I believe Randall et al 2007 first coined the phrase "EPP-IE". They should be cited on line 8 of page 3. They already are cited elsewhere.*

Thanks for pointing this out, the reference to Randall et al. (2007) has been added at this point.

*Siskind references get the first initials wrong. In one place its DR in another its DW. Should be DE*

Changed.

*Same problem with Fleimng- its EK in the reference with Jackman. EL in other places.*

Changed to the correct (EL).

*Matthes paper is 2017 as it has appeared now.*

Changed.

**Reply to interactive comment 1**

*The paper states that "Analysis of several decades of re-analysis data show a warming of the mid-to late winter upper stratosphere related to high geomagnetic activity (Lu et al., 2008; Seppälä et al., 2013). These have been interpreted as a result of coupling between the vortex strength and wave propagation and reflection, an assumption strengthened by the apparent relation to the phase of the stratospheric quasi-biennal oscillation and the solar cycle (Lu et al., 2008; Seppälä et al., 2013). However, our results suggest that the direct radiative impact plays a role as well."*

*As we write (about the upper stratospheric warming signal) in (Seppälä et al., 2013), the paper referenced here: Based on earlier work by others, Baumgaertner et al. (2011) suggested that the warming signal would be a result in decrease in ozone radiative cooling as a response to ozone depletion, and the cooling signal might arise from dynamical heating due to slowing down of the meridional Brewer-Dobson circulation. Such a reduction would be associated with less upward EP flux and more waves reflecting toward the equator [see (Lu et al., 2008b), and references therein]. As discussed above, this is now confirmed by our EP flux results. (Discussion, last paragraph)*

*These results confirm the previous hypothesis of (Lu et al., 2008b) regarding the role of dynamics in coupling geomagnetic activity levels and stratospheric changes and supports the suggestion of (Baumgaertner et al., 2011) about the dynamical coupling mechanism connecting EPP-NOx induced ozone loss, polar stratospheric temperatures, and the modulation of the Northern Annular Mode. (Conclusions, last paragrapaph)*

*This clearly shows that the role of the direct radiative impact in the upper stratosphere suggested here is in no disagreement by the works cited. Results of (Lu et al., 2008) and (Seppälä et al., 2013) both support the suggestion of (Baumgaertner et al., 2011) (which is unfortunately not cited in the current paper in this context) that the upper stratosphere warming signal is related to the radiative changes arising from changes in ozone, exactly as proposed again here.*

*In summary, the results of the modelling work done here regarding the role of direct radiative impact do, in fact, strongly support the results of (Lu et al., 2008; Baumgaertner et al., 2011; Seppälä et al., 2013). This agreement between the independent studies is important, and I would be grateful if the authors would correct their statement concerning this in the current paper.*

Thanks for pointing this out. We have included a reference to Baumgaertner et al. (2011) in the Conclusions and in the Introduction. The text in the Conclusions has been changed to

[revised manuscript text omitted]

**List of changes**

**Substantial changes**

- Changed Pa to hPa in the text as well as in all figures and tables.

- Corrected bug in zonal averaging of 3dCTM data (small changes in Figures 2, 4, and 6)

- Changed title of Section 4: *Ozone intercomparison, quantification of ozone loss, and net radiative heating rate change*

- Included new Subsection 4.1: *Comparison of modeled and observed ozone fields*

- Included new Figures 8 and 9 in Subsection 4.1 (Comparison of ozone at high Southern and Northern latitudes to MIPAS data

- Included new Subsection 4.2: *Comparison of modeled and observed ozone anomalies*

- Included new Figures 10 and 11 in Subsection 4.2 (relative anomalies of MIPAS compared to models in high Southern andNorthern latitudes)

- Changed title of former Subsection 4.1 (now Subsection 4.3) to *Modeled ozone anomalies due to particle precipitation*

- Included contour lines showing absolute values of ozone change (ppm) in Figures 12 and 13 (former Figures 8 and 9)

- Included discussion of absolute ozone differences in Subsection 4.3 (former Subsection 4.1): *Absolute differences in the stratospheric winter-time ozone anomalies are largest in EMAC (more than 0.5 ppm in all winters, more than 1 ppm in all winters but 2009, more than 1.5 ppm in winters 2003 and 2005 between 10–1 hPa), lower in 3dCTM (0.5–1 ppm in all winters but 2009), lowest in KASIMA (more than 0.5 ppm only in winter 2003). Values for winter 2003 are in good agreement with a previous model study incorporating MIPAS NOx in the lower mesosphere and showing 1–1.5 ppm ozone loss between 30–40 km compared to a model run without excess NO2 (Reddmann et al., 2010).*

- Included discussion of solar proton event impact on ozone changes in Subsection 4.3 (former Subsection 4.1): *Strong solar proton events in October 2003, January 2005 and December 2006 lead to an instantaneous loss of ozone from the upper stratosphere to the mesopause (10–0.01 hPa). In the mesosphere, this impact is shortlived and restricted to a few days only, while in the stratosphere below 1 hPa, ozone loss continues throughout the summer.*

- Included comparison of mesospheric ozone loss to CMAM results (page 34, lines 7–11)

- Included comparison of stratospheric ozone loss to results from previous studies (Natarajan et al., 2004; Randall et al., 2005; Reddmann et al., 2010; Semeniuk et al., 2012), on page 35, lines 10–21 of the revised manuscript

- Included net radiative heating rate (the sum of the shortwave and longwave contributions) from the EMAC models in Figures 14 and 15 (former Figures 10 and 11)

- Included discussion of relative heating rate changes, page 37, line 23 to page 40, line 2 of the revised manuscript

- Included in the conclusion: *which is also reproduced in model experiments using free-running chemistry-climate models (Semeniuk et al., 2011; Baumgaertner et al., 2011). Based on older model experiments by (Langematz et al., 2005), Baumgaertner et al. (2011) and Seppälä et al. (2013) argue that the warming in the upper stratosphere and lower mesosphere is consistent with a direct radiative impact, while a cooling of the middle and lower stratosphere observed at the same time, during mid-winter (DJF in the Northern hemisphere), is more likely the result of coupling between the vortex strength and wave propagation and reflection, an assumption strengthened by the apparent relation to the phase of the stratospheric quasi-biennal oscillation and the solar cycle (Lu et al., 2008; Seppälä et al., 2013). Our results are consistent with these earlier studies, and strengthen the assumption that the mid-winter warming is due at least partly to a direct radiative impact*, page 41 lines 15–20 of the revised manuscript

**Minor changes**

– Introduction: inserted *and in events with particularly hard energy spectra even down to the lower stratosphere* (page 2 line 14-15 of the revised manuscript)

– Inserted reference to *Randall et al., 2007* in the Introduction (page 3, line 8 of the revised manuscript)

– Inserted *Lower values of ozone in the upper stratosphere are observed during winters characterized by large particle forcing or enhanced values of NOy (Natarajan et al., 2004; Randall et al., 2005),* in the Introduction (page 3, lines 10–11)

– Inserted *or a model experiment with freely adaptable dynamics was carried out, making comparison to observations more difficult.* in the Introduction (page 3, line 21 of the revised manuscript)

– Inserted *ozone intercomparisons with MIPAS data* in the Introduction (page 4, line 5 of the revised manuscript)

– Adapted sentence on page 5, lines 33–34 of the revised manuscript (KASIMA description)

– Page 7, line 20 of the revised manuscript: included $O_3$ in list of species used from MIPAS data

– Included *Updates in the ozone retrieval scheme since von Clarmann et al. (2013) are documented in Laeng et al. (2014)* in the first line of page 8 of the revised manuscript

– Included *The thin black contours refer to -1.5 (solid), -1 (dashed) and -0.5 (solid) ppm* in caption of Figures 12 and 13 (former Figures 8 and 9)

– Included *changing the partitioning of HOx from HO2 to OH* and reference to Verronen and Lehmann, 2015, on page 34, line 14 of the revised manuscript

– Included *and NO + HO2 ⟶ NO2 + OH are also processed, changing the partitioning of HOx and destroying larger amounts of ozone than in the Base run* on page 34, line 20–21, of the revised manuscript

– Included *Solar proton events in October 2003, January 2005 and September 2005 lead to instantaneous ozone loss from the mid-stratosphere to the upper mesosphere in all models.* in the discussion of Northern hemisphere ozone loss on page 34, lines 28–29, of the revised manuscript

[revised manuscript text omitted]

---

## Author Response (AR2)

**Reply to reviewer 1**

We would like to thank the reviewer for reviewing the paper a second time, and for her or his careful review and useful comments.

*Fundamentally, I also concur with their fundamental point that the SH winter sees continuous downwelling while the NH is dynamically controlled. However, I would quibble with their terminology slightly- stratospheric warmings can be minor or major, but "strong" isn't really a standard descriptor (see the paper by Butler et al in the Bulletin of the AMS, 2015, page 1913). Rather the events in 2004, 2006 and 2009 were characterized by prolonged or extended (depending upon which paper you read) recoveries with elevated stratopauses that are linked to enhanced descent. The present paper is a bit muddled in how they describe it. I am reluctant to suggest that they add too many more references; however I can suggest Randall et al [GRL, 2006] for the the 2006 event. (They have Manney, but she didn't discuss the NOx descent the way Randall does). Papers by Siskind (2007, 2010) or Chandran (2010, 2012) or Limpasuvan (2016) discuss the elevated stratopause phenomena. They may wish to add some citations to those events because they do seem to be different than the conventional major stratospheric warming.*

Thanks for pointing this out. Five references were included: Randall et al. (2006,2009), Siskind et al. (2007), Orsolini et al. (2010) and Limpasuvan et al. (2016). The text in the introduction (first paragraph of page 3) was changed as follows:

"Particularly high values of EPP NOx are observed in Northern hemisphere late winters after the strong sudden stratospheric warming events in winters 2003/2004, 2005/2006, and 2008/2009 (Randall et al., 2006; 2009; Funke et al., 2014a; Sinnhuber et al., 2014; Funke et al., 2017). These warmings were followed by long-lasting downwelling in the mesosphere and upper stratosphere enabled by a strong polar vortex re-forming after the event. It was shown both from observations (Siskind et al. 2007; Orsolini et al., 2010) and model results (Limpasuvan et al., 2016) that this period of enhanced downwelling was characterized by the formation of an elevated stratopause in the upper mesosphere."

In the description of the dynamical state of the atmosphere in the model period, the following text was added (line 19 of page 10):

"These warmings were followed by an elevated stratopause and strong and long-lasting mesospheric and upper stratospheric descent (Randall et al., 2006; 2009; Orsolini et al., 2010; Limpasuvan et al., 2016). They will be called *strong sudden stratospheric warmings* in the following."

*Finally, regarding summer events in the NH, Siskind (ACP, 2016) talked about how the consequences of dynamical variability of the previous winter could get frozen into the summer and lead to lower ozone- without particle precipitation. I wonder if this is relevant.*

This is certainly relevant for stratospheric ozone during summer, but not in the context of this paper particularly, as we derived ozone loss from the chemical changes in the models only by comparing nudged model runs with and without particle forcing.

*They do have a lot of words and perhaps get tripped up in the details. For example, at the bottom of page 34 (line 33) they start a sentence "in the solar maximum winters" and then, list a whole bunch of years in parentheses including 2005-2006 and 2008-2009 (top of page 35). Those years are hardly solar maximum. They should clean this up a bit.*

Again, thanks for pointing this out. In the sentence starting on page 34, line 33, the mention of winters 2005/2006 and 2008/2009 was erased. Ozone loss in these winters is discussed again later (line 5 of page 35), and this sentence was changed slightly:

"In EMAC, downwelling ozone losses of 10–30% are also observed outside solar maximum related to the sudden stratospheric warmings in early 2006 and early 2009; these are weaker in KASIMA, and not observed in 3dCTM."

*Another example of problematic presentation was that in general, I found Figures 14 and 15 hard to read and the text – particularly the new stuff on page 37, often confused me. They should consider cutting the plots off at 10 hpa, rather than going down to the troposphere.*

Figures 14 and 15 are now cut at 30 hPa.

*As far as wording problems, for example, last line of page 37. "net radiative rate of 1-2.5K" ? Do they mean K/day?*
Yes. This was corrected.

*Also, Line 27 on page 37, "net values" – is that "net changes" or "net absolute values". And that whole sentence is garbled- they say "in the range 1-2.5 K/day.......... reaching more than 0.5 K/day........". Well, 0.5K/day is not a "reach" if its in*

*the middle of their range. Or alternatively, if the 1-2.5 K/day is the absolute value, then a change of 0.5 K/day is at least 20 % of the absolute value, not 2-10 % as stated on line 29. So no matter how I look at the discussion, I find it confusing.*

Thanks for pointing this out. The "net values" are indeed the "The net radiative heating rates". Absolute values in the range 10-1 hPa are in the range 1–10 K/day, so the value given in the text was wrong. This has been corrected as follows:

5 "The net radiative heating rates increase from 1 K/day at 10 hPa to more than 10 K/day at 1 hPa. Changes due to particle precipitation are in the order of magnitude of 0.1 K/day in this altitude range, reaching more than 0.5 K/day in Southern hemisphere winter 2003 (EMAC). This amounts to relative changes of 1–10 % in most winters, up to 50 % in winter 2003 (EMAC)."

*Small reference glitch: I could not find a citation in the text to Prather, 1986, which is given in their reference list.*

10 Thanks for pointing this out. The reference was erased from the list.

**List of changes**

- Included references to Randall et al., 2006; Randall et al., 2009; Siskind et al., 2007; Orsolini et al., 2012; Limpasuvan et al., 2016.

- Inserted "2005/2006" at line 6 of page 3 in the adapted manuscript.

- Inserted reference to Randall et al., 2006 and Randall et al., 2009 in line 6 of page 3 in the adapted manuscript.

- Inserted "These warmings were followed by long-lasting downwelling in the mesosphere and upper stratosphere enabled by a strong polar vortex re-forming after the event. It was shown both from observations (Siskind et al., 2007; Orsolini et al., 2010) and model results (Limpasuvan et al., 2016) that this period of enhanced downwelling was characterized by
10
  the formation of an elevated stratopause in the upper mesosphere." at line 7 of page 3 in the adapted manuscript.

- Inserted "These warmings were followed by an elevated stratopause and strong and long-lasting mesospheric and upper stratospheric descent (Randall et al., 2006, 2009; Orsolini et al., 2010; Limpasuvan et al., 2016). They will be called strong sudden stratospheric warmings in the following" at lines 19–21 of page 10 in the adapted manuscript.

- Erased "2005-2006" and "2008-2009" on line 35 of page 34 of the adapted manuscript.

- Inserted of "10–30%" and "outside solar maximum" at line 5 of page 35 of the adapted manuscript.

- Changed sentence starting on line 28 of page 37 of the adapted manuscript to "The net radiative heating rates increase from 1 K/day at 10 hPa to more than 10 K/day at 1 hPa. Changes due to particle precipitation are in the order of magnitude of 0.1 K/day in this altitude range, reaching more than 0.5 K/day in Southern hemisphere winter 2003 (EMAC). This amounts to relative changes of 1–10 % in most winters, up to 50 % in winter 2003 (EMAC)."

- Inserted "0.1 K/day in" at line 34 of page 37 of the adapted manuscript.

- Cut Figures 14 and 15 at 30 hPa.

- Changed "K" to "K/day" in line 1 of page 40 of the adapted manuscript.

[revised manuscript text omitted]